# LEMMA-RCA: A Large Multi-modal Multi-domain Dataset for Root Cause Analysis

## Abstract

Root cause analysis (RCA) is crucial for enhancing the reliability and performance of complex systems. However, progress in this field has been hindered by the lack of large-scale, open-source datasets tailored for RCA. To bridge this gap, we introduce LEMMA-RCA, a large dataset designed for diverse RCA tasks across multiple domains and modalities. LEMMA-RCA features various real-world fault scenarios from Information Technology (IT) and Operational Technology (OT) systems, encompassing microservices, water distribution, and water treatment systems, with hundreds of system entities involved. We evaluate the performance of fourteen baseline methods on LEMMA-RCA across various settings, including offline and online modes, as well as single and multi-modal configurations. The dataset is publicly available at `https://lemma-rca.github.io/`.

## 1 Introduction

Root cause analysis (RCA) is essential for identifying the underlying causes of system failures, ensuring the reliability and robustness of real-world systems. Recent advancements in artificial intelligence and software development have led to increased complexity and interdependence in modern systems. This complexity heightens their vulnerability to faults arising from interactions among modular services, which can disrupt user experiences and incur significant financial losses. Traditional manual RCA, however, is labor-intensive, costly, and prone to errors due to the complexity of systems and the extensive data involved. Therefore, efficient and effective data-driven RCA methods are crucial for pinpointing failures and mitigating financial losses when system faults occur.

Root cause analysis has been extensively studied across various domains and settings (Capozzoli et al., 2015; Deng & Hooi, 2021; Brandón et al., 2020; Fourlas & Karras, 2021; Gao et al., 2015). Based on the application scenarios, RCA can be carried out in offline/online fashion with single/multi-modal system data. Existing studies on RCA in these settings involve numerous learning techniques such as Bayesian methods (Alaeddini & Dogan, 2011), decision trees (Chen et al., 2004), *etc*. Particularly, causal structure learning based technique (Burr, 2003; Pamfil et al., 2020; Ng et al., 2020; Tank et al., 2022; Yu et al., 2023; Wang et al., 2023a;b; Zheng et al., 2024) has proven effective in constructing causal or dependency graphs between different system entities and key performance indicators (KPIs), thereby enabling the tracing of underlying causes through these structures.

Data is the oxygen of data-driven methods. Despite significant progress in RCA techniques, the availability of large-scale public datasets remains limited, often due to confidentiality concerns (Harsh et al., 2023). This scarcity hinders fair comparisons between RCA methods. Additionally, publicly accessible datasets often contain manually injected faults rather than real faults, and each dataset typically covers only a single domain. These limitations can prevent existing RCA methods from effectively identifying various types of system faults in real-world scenarios, potentially leading to regulatory and ethical consequences in critical sectors.

To address these limitations, we introduce **LEMMA-RCA**, a collection of Large-scalE Multi-ModAl datasets with various real system faults to facilitate future research in Root Cause Analysis. LEMMA-RCA encompasses real-world applications such as IT operations and water treatment systems, with **hundreds of system entities** involved. LEMMA-RCA accommodates **multi-modal** data including textual system logs with millions of event records and time series metric data with more than $100,000$ timestamps. We annotate LEMMA-RCA with ground truth labels indicating the precise time stamps when **real system faults** occur and their corresponding root-cause system entities.

Table 1: **Existing datasets for root cause analysis.** The top row corresponds to our dataset. The symbols ✓ and ✗ indicate that the dataset has or does not have the corresponding feature, respectively.

| Dataset | Public | Real Faults | Large-scale | Multi-domain | Dependency Graph | Modality | |
|---|---|---|---|---|---|---|---|
| | | | | | | Single | Multiple |
| LEMMA-RCA | ✓ | ✓ | ✓ | ✓ | ✓ | ✓ | ✓ |
| NeZha | ✓ | ✗ | ✗ | ✗ | ✗ | ✓ | ✓ |
| PetShop | ✓ | ✗ | ✗ | ✗ | ✓ | ✓ | ✗ |
| Sock-Shop | ✗ | ✗ | ✗ | ✗ | ✗ | ✓ | ✗ |
| ITOps | ✗ | ✓ | ✓ | ✗ | ✗ | ✓ | ✗ |
| Murphy | ✗ | ✓ | ✗ | ✗ | ✗ | ✓ | ✗ |

A comparison between LEMMA-RCA and existing datasets for RCA is presented in Table 1. We briefly discuss the status of existing datasets: 1) *NeZha* (Yu et al., 2023) has limited size and contains many missing parts in the monitoring data, and it is confined to one domain: microservice architectures. 2) *PetShop* (Saurabh Garg, Imaya Kumar Jagannathan, 2024) has a small size. Additionally, the system comprises only 41 components, limiting its complexity and reducing the practicality for real-world scenarios. 3) *Sock-Shop* (Ikram et al., 2022) is small-scale with only 13 microservices, and the injected faults (CPU hog and memory leak) are synthetic. Additionally, the data is not publicly available and consists solely of single-modality metrics, lacking diversity in data sources such as logs or traces. 4) *ITOps* (Li et al., 2022c) dataset is not public and contains structured logs that do not contribute to comprehending the underlying causal mechanism of system failures, making it difficult to conduct fine-grained RCA. 5) *Murphy* (Harsh et al., 2023) is collected from a simple system and also not public. In comparison to prior work, LEMMA-RCA demonstrates a comprehensive maturity on the accessibility, authenticity, and diversity.

LEMMA-RCA enables fair comparisons among different RCA methods. We evaluate fourteen baseline methods, with eleven suited for offline settings and the remaining three designed for online RCA. The quality of various data modalities is assessed in both online and offline setups. The experimental results demonstrate the effectiveness of LEMMA-RCA on evaluating related methods and its extensive utility for advanced research in root cause analysis.

## 2 PRELIMINARIES AND RELATED WORK

**Key Performance Indicator (KPI)** is a monitoring time series that indicates the system status. For instance, latency and service response time are two common KPIs used in microservice systems. A large value of latency or response time usually indicates a low-quality system performance or even a system failure.

**Entity Metrics** are multivariate time series collected by monitoring numerous system entities or components. For example, in a microservice system, a system entity can be a physical machine, container, pod, *etc*. Some common entity metrics in a microservice system include CPU utilization, Memory utilization, disk IO utilization, *etc*. An abnormal system entity is usually a potential root cause of a system failure.

**Data-driven Root Cause Analysis Problem**. Given the monitoring data (including metrics and logs) of system entities and system KPIs, the root cause analysis problem is to identify the top $K$ system entities that are most relevant to KPIs when a system fault occurs. RCA techniques can be implemented in various settings, where offline/online and single-modal/multi-modal are mostly commonly concerned. Offline RCA is conducted retrospectively with historical data to determine past failures, whereas online RCA operates in real-time using current data streams to promptly address issues. On the other hand, single-modal RCA relies solely on one type of data for a focused analysis, while multi-modal RCA uses multiple data sources for a comprehensive assessment. We illustrate the procedure of RCA in single-modal offline and multi-modal online settings in Figure 1.

**Single-modal Offline Root Cause Analysis (RCA)** retrospectively identifies the primary cause of system failures using a single data type after an event has occurred (Wang et al., 2023b; Tang et al., 2019; Meng et al., 2020b; Li et al., 2021; Soldani & Brogi, 2022). For example, Meng *et al.* (Meng et al., 2020b) analyze monitoring metric data to discern sequential relationships and integrate causal and temporal information for root cause localization in microservice systems. Similarly, Wang *et*

*al.* (Wang et al., 2023b) construct an interdependent causal network from time series data, using a random walk strategy to pinpoint the most probable root causes. Li *et al.* (Li et al., 2021) evaluate microservice traces, determining that a service with a higher ratio of abnormal to normal traces is likely the root cause. Recently, large language model (LLM) based methods become a new research direction to learn causal relation for root cause identification due to the success of LLMs in performing complex tasks (Chen et al., 2024; Shan et al., 2024; Goel et al., 2024; Zhou et al., 2024; Roy et al., 2024; Wang et al., 2024). For instance, Chen *et al.* (Chen et al., 2024) introduce RCACopilot, an innovative on-call system empowered by the large language model for automating RCA of cloud incidents. Shan *et al.* (Shan et al., 2024) propose to first identify the log messages indicating configuration-related errors and then localize the suspected root-cause configuration properties based on the selected log messages and the offered configuration settings by LLMs. Goel *et al.* (Goel et al., 2024) demonstrate that LLMs can benefit from service functionality and upstream dependency information in better reasoning, thus improving the quality of the identification of root causes. Although these studies demonstrate notable efficacy, they rely exclusively on single-modal data, which may lead to suboptimal and biased outcomes in root cause localization.

**Multi-modal Offline RCA**. Recent studies have explored utilizing multi-modal data for offline RCA, which can be divided into two approaches (Yu et al., 2023; Hou et al., 2021; Zheng et al., 2024; Lan et al., 2023). The first approach, exemplified by Nezha (Yu et al., 2023) and PDiagnose (Hou et al., 2021), involves extracting information from each modality separately and then integrating it for analysis. Conversely, the second approach focuses on the interactions between different modalities. For instance, MULAN (Zheng et al., 2024) develops a comprehensive causal graph by learning correlations between modalities, while MM-DAG (Lan et al., 2023) aims to jointly learn multiple Direct Acyclic Graphs, improving both consistency and depth of analysis.

**Online RCA**. Despite significant advances, most RCA methods are designed for offline use, requiring extensive data collection and full retraining for new faults, which delays response times. To address this, Wang *et al.* (Wang et al., 2023a) introduced an online RCA method that decouples state-invariant and state-dependent information and incrementally updates the causal graph. Li *et al.* (Li et al., 2022a) developed a causal Bayesian network that leverages system architecture knowledge to mitigate potential biases toward new data. However, these methods are limited to single-modal data, and there is a critical need for online RCA methods that can effectively handle multi-modal data

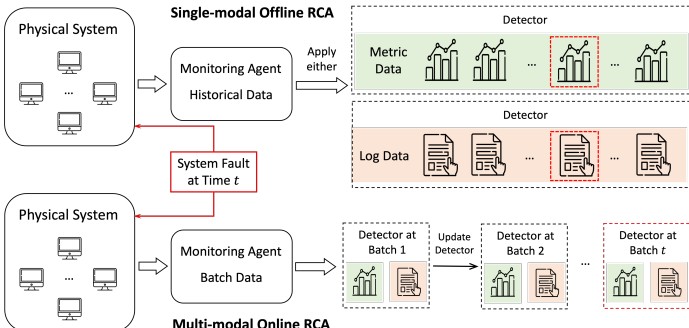

Figure 1: Illustration of RCA workflow in the single-modal offline setting (top) and the multi-modal online setting (bottom). The other two settings can be viewed as an ensemble of corresponding components (data collection, detector, modality) and follow the same systematic procedure.

## 3 LEMMA-RCA Data

This section outlines the data resources, details the preprocessing steps, and presents visualizations to illustrate the characteristics of the data released. The data licence can be found in appendix D.

### 3.1 Data Collection

We collect real-world data from two domains: IT operations and OT operations. The IT domain includes sub-datasets from Product Review and Cloud Computing microservice systems, while the

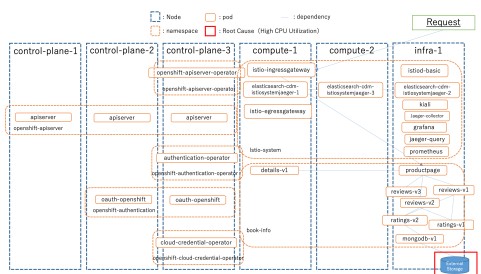 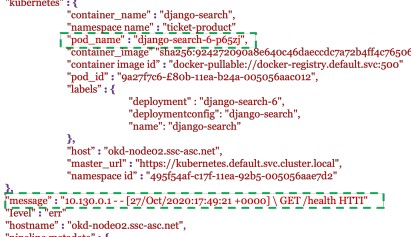

(a) The architecture of Product Review Platform     (b) Log data captured by the ElasticSearch

Figure 2: Visualization of the microservice system platform, which contains 6 nodes and multiple pods that may vary across different stages; and the ElasticSearch log data.

OT domain includes Secure Water Treatment (SWaT) and Water Distribution (WADI) sub-datasets from water treatment and distribution systems. Data specifics are provided in Table 2.

In the IT domain, we developed two microservice platforms: the **Product Review** Platform and the **Cloud Computing** Platform. The Product Review Platform is composed of six OpenShift nodes (such as ocp4-control-plane-1 through ocp4-control-plane-3, ocp4-compute-1 and ocp4-compute-2, and ocp4-infra-1) and 216 system pods (including ProductPage, MongoDB, review, rating, payment, Catalogue, shipping, *etc*.). In this setup, four distinct system faults are colleceted, including out-of-memory, high-CPU-usage, external-storage-full, and DDoS attack, on four different dates. Each system fault ran the microservice system for at least 49 hours with different pods involved. The pods running in different stages may vary, and the pods associated with different types of system faults also differ. The structure of this microservice system with some key pods during one fault is depicted in Figure 2 (a). Both log and metric data were generated and stored systematically to ensure comprehensive monitoring. Specifically, eleven types of node-level metrics (*e.g.*, net disk IO usage, net disk space usage, *etc*.) and six types of pod-level metrics (*e.g.*, CPU usage, memory usage, *etc*.) were recorded by Prometheus (Turnbull, 2018), and the time granularity of these system metrics is 1 second. Log data, on the other hand, were collected by ElasticSearch (Zamfir et al., 2019) and stored in JSON files with detailed timestamps and retrieval periods. The contents of system logs include timestamp, pod name, log message, *etc*., as shown in Figure 2 (b). The JMeter (Nevedrov, 2006) was employed to collect the system status information, such as elapsed time, latency, connect time, thread name, throughput, *etc*. The latency is considered as system KPI as the system failure would result in the latency significantly increasing.

For the Cloud Computing Platform, we monitored six different types of faults (such as cryptojacking, mistakes made by GitOps, configuration change failure, *etc*.), and collected system metrics and logs from various sources. In contrast to the Product Review platform, system metrics were directly extracted from CloudWatch[1] Metrics on EC2 instances, and the time granularity of these system metrics is 1 second. Log events were acquired from CloudWatch Logs, consisting of three data types (*i.e.*, log messages, api debug log, and mysql log). Log message describes general log message about all system entities; api debug log contains debug information of the AP layer when the API was executed; mysql logs contain log information from database layer, including connection logs to mysql, which user connected from which host, and what queries were executed. Latency, error rate, and utilization rate were tracked using JMeter tool, serving as Key performance indicators (KPIs). This comprehensive logging and data storage setup facilitates detailed monitoring and analysis of the system's performance and behavior.

In the OT domain, we constructed two sub-datasets, SWaT and WADI, using monitoring data collected by the iTrust lab at the Singapore University of Technology and Design (iTrust, 2022). These two sub-datasets consist of time-series/metrics data, capturing the monitoring status of each sensor/actuator as well as the overall system at each second. Specifically, SWaT (Mathur & Tippenhauer, 2016) was collected over an 11-day period from a water treatment testbed equipped with 51 sensors. The system operated normally during the first 7 days, followed by attacks over the last 4 days, resulting in 16 system faults. Similarly, WADI (Ahmed et al., 2017) was gathered from a water distribution testbed

---

[1]https://aws.amazon.com/cloudwatch/

Table 2: Data statistics of IT and OT operation sub-datasets.

| Microservice System (IT) | Product Review | Cloud Computing |
|---|---|---|
| Original Dataset Size | 765 GB | 540 GB |
| Number of (#) fault types | 4 | 6 |
| Average # entities per fault | 216.0 | 167.71 |
| Average # metrics per fault | 11 (node-level) + 6 (pod-level) | 6 (node-level) + 7 (pod-level) |
| Average # timestamps per fault | 131,329.25 | 109,350.57 |
| Average max log events per fault across pods | 153,081,219.0 | 63,768,587.25 |
| Water Treatment/Distribution (OT) | SWaT | WADI |
| Original Dataset Size | 4.47G | 5.67G |
| Number of (#)fault types | 16 | 9 |
| Average # entities per fault | 51.0 | 123.0 |
| Average # metrics per fault | 7 (node-level) + 7 (pod-level) | 7 (node-level) + 7 (pod-level) |
| Average # timestamps per fault | 56239.88 | 85248.47 |

over 16 days, featuring 123 sensors and actuators. The system maintained normal operations for the first 14 days before experiencing attacks in the final 2 days, with 15 system faults recorded.

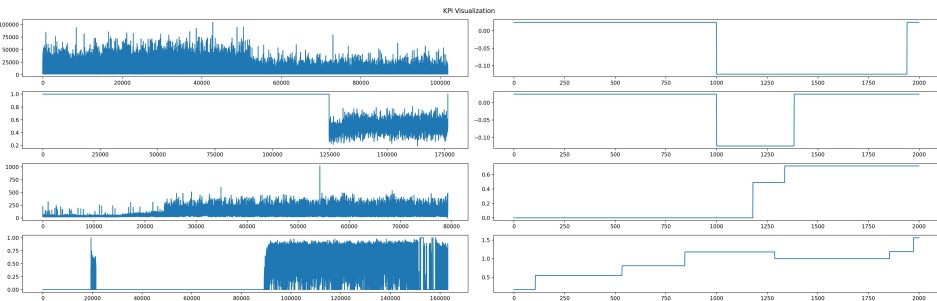

Figure 3: Visualization of KPI for system failure cases. **Left**: the first two sub-figures are from the Product Review sub-dataset; the third and fourth sub-figures are from the Cloud Computing sub-dataset; **Right**: the first two sub-figures are from the SWaT sub-dataset; the last two sub-figures are from the WADI sub-dataset.

We visualized the key performance indicator (KPI) for eight failure cases in Figure 3, where sudden spikes or drops in latency indicate system failures. The first two sub-figures on the left show the KPIs for two faults in the Product Review sub-dataset, while the third and fourth sub-figures depict faults in the Cloud Computing sub-dataset. The first two sub-figures on the right display faults in the SWaT dataset, and the last two show faults in the WADI dataset. The x-axis represents the timestamp, and the y-axis shows the system latency.

## 3.2 DATA PREPROCESSING

After collecting system metrics and logs, we assess whether each pod exhibits stationarity, as non-stationary data are unpredictable and cannot be effectively modeled. Consequently, we exclude non-stationary pods, retaining only stationary ones for subsequent data preprocessing steps.

**Log Feature Extraction for Product Review and Cloud Computing.** The logs of some system entities we collected are limited and insufficient for meaningful root cause analysis. Thus, we exclude them from further analysis. Additionally, the log data is unstructured and frequently uses a special token, complicating its direct application for analysis. How to extract useful information from unstructured log data remains a great challenge. Following (Zheng et al., 2024), we preprocess the log data into time-series format. We first utilize a log parsing tool, such as Drain, to transform unstructured logs into structured log messages represented as templates. We then segment the data using fixed 10-minute windows with 30-second intervals, calculating the occurrence frequency of each log template. This frequency forms our first feature type, denoted as $X_1^L \in \mathbb{R}^T$, where $T$ is the number of timestamps. We prioritize this feature because frequent log templates often indicate critical insights, particularly useful in identifying anomalies such as Distributed Denial of Service (DDoS) attacks, where a surge in template frequency can indicate unusual activity.

Moreover, we introduce a second feature type based on 'golden signals' derived from domain knowledge, emphasizing the frequency of abnormal logs associated with system failures like DDoS

attacks, storage failures, and resource over-utilization. Identifying specific keywords like 'error,' 'exception,' and 'critical' within log templates helps pinpoint anomalies. This feature, denoted as $X_2^L \in \mathbb{R}^T$, assesses the presence of abnormal log templates to provide essential labeling information for anomaly detection.

Lastly, we implement a TF-IDF based method, segmenting logs using the same time windows and applying Principal Component Analysis (PCA) to reduce feature dimensionality, selecting the most significant component as $X_3^L \in \mathbb{R}^T$. We concatenate these three feature types to form the final feature matrix $X^L = [X_1^L; X_2^L; X_3^L] \in \mathbb{R}^{3 \times T}$, enhancing our capacity for a comprehensive analysis of system logs and improving anomaly detection capabilities.

**KPI Construction for SWaT and WADI**. The SWaT and WADI sub-datasets include the label column that reflects the system status; however, the values within this column are discrete. To facilitate the root cause analysis, it is beneficial to transform these values into a continuous format. Specifically, we propose to convert the label into a continuous time series. To achieve this, we employ anomaly detection algorithms, such as Support Vector Data Description and Isolation Forest, to model the data. Subsequently, the anomaly score, as determined by the model, will be utilized as the system KPI. More data preprocessing details on SWaT and WADI can be found in Appendix A

### 3.3 System Fault Scenarios

There are 10 different types of real system faults in Product Review and Cloud Computing sub-datasets. Due to the space limitation, we select two representative cases (one from each) and provide the details below. Other fault scenarios are presented in Appendix B. We also visualize the system fault of these two cases in Figure 4.

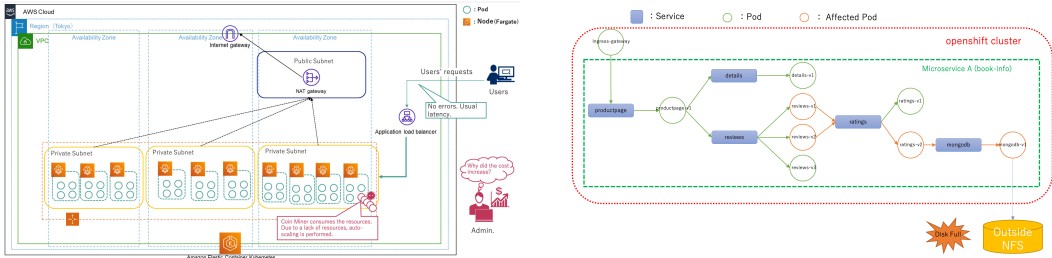

Figure 4: Visualization of two system fault scenarios. **Left:** Cryptojacking. **Right:** External storage failure.

- **Cryptojacking**. In this scenario, cloud usage fees increase due to cryptojacking, where a Coin Miner is covertly downloaded and installed on a microservice (details-v1 pod) in an EKS cluster. This miner gradually consumes IT resources, escalating the cloud computing costs. Identifying the root cause is challenging because the cost (SLI) encompasses the entire system, and no individual service errors are detected. Periodic external requests are sent to microservices, and after a day, the miner's activity triggers auto-scaling in details-v1, increasing resource usage. Fargate's impact on EKS costs is significant due to its resource dependency. KPI (SLI) is calculated from resource usage, with all pod and node metrics collected from CloudWatch. However, there are no node logs for Fargate, complicating diagnosis.

- **External Storage Failure**. In this system failure, we fill up the external storage disk connected to the Database (DB) pod (*i.e.*, mongodb-v1) within Microservice A's OpenShift[2] cluster. When the storage becomes full, the DB pod cannot add new data, resulting in system errors. These errors propagate to pods that depend on the DB pod, causing some services (ratings) within Microservice A to encounter errors. We monitor changes in response and error information for Microservice A using Jaeger logs. Metrics for all containers and nodes, including CPU and memory usage, are obtained from Prometheus within OpenShift. Logs for all containers and nodes are retrieved from Elasticsearch within OpenShift. Additionally, we collect message logs from the external storage. We illustrate the metrics and log data of the root cause pod in Figure 5.

---

[2]https://www.redhat.com/en/technologies/cloud-computing/openshift

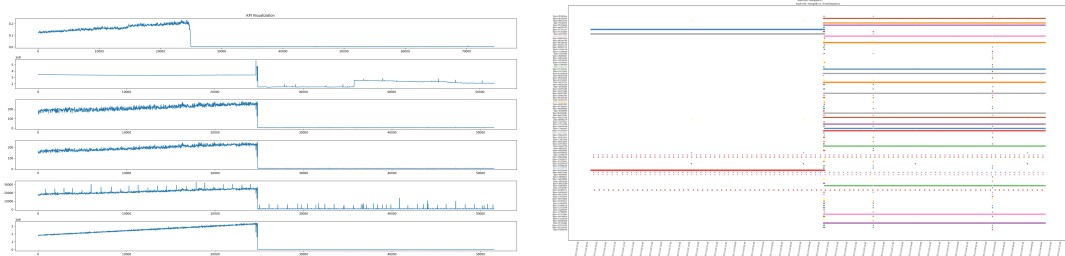

Figure 5: Visualization of root cause for one system failure case (*i.e.*, **External Storage Failure**) on the Product Review Platform. **Left:** six system metrics of root cause. **Right:** the system log of the root cause pod (*i.e.*, Mongodb-v1) with the x-axis representing the timestamp, the y-axis indicating the log event ID, and the colored dots denoting event occurrences. Sudden drops in the metrics data, as well as new log event patterns observed at the midpoint, indicate a system failure.

## 4 EXPERIMENTS

### 4.1 EXPERIMENTAL SETUP

**Evaluation Metrics**. To assess baseline RCA method on LEMMA-RCA, we choose three widely-used metrics (Wang et al., 2023b; Meng et al., 2020a; Zheng et al., 2024) and introduce them below.

(1). **Precision@K (PR@K)**: It measures the probability that the top $K$ predicted root causes are real, defined as:

$$\text{PR@K} = \frac{1}{|\mathbb{A}|} \sum_{a \in \mathbb{A}} \frac{\sum_{i<k} R_a(i) \in V_a}{\min(K, |v_a|)} \tag{1}$$

where $\mathbb{A}$ is the set of system faults, $a$ is one fault in $\mathbb{A}$, $V_a$ is the real root causes of $a$, $R_a$ is the predicted root causes of $a$, and $i$ is the $i$-th predicted cause of $R_a$.

(2). **Mean Average Precision@K (MAP@K)**: It assesses the top $K$ predicted causes from the overall perspective, defined as:

$$\text{MAP@K} = \frac{1}{K|\mathbb{A}|} \sum_{a \in \mathbb{A}} \sum_{i \le j \le K} \text{PR@j} \tag{2}$$

where a higher value indicates better performance.

(3). **Mean Reciprocal Rank (MRR)**: It evaluates the ranking capability of models, defined as:

$$\text{MRR@K} = \frac{1}{|\mathbb{A}|} \sum_{a \in \mathbb{A}} \frac{1}{\text{rank}_{R_a}} \tag{3}$$

where $rank_{R_a}$ is the rank number of the first correctly predicted root cause for system fault $a$.

**Baselines**. We evaluate the performance of the following RCA models on the benchmark sub-datasets: (1). **PC** (Burr, 2003): This classic constraint-based causal discovery algorithm is designed to identify the causal graph's skeleton using an independence test. (2) **Dynotears** (Pamfil et al., 2020): It constructs dynamic Bayesian networks through vector autoregression models. (3). **C-LSTM** (Tank et al., 2022): This model utilizes LSTM to model temporal dependencies and capture nonlinear Granger causality. (4). **GOLEM** (Ng et al., 2020): GOLEM relaxes the hard Directed Acyclic Graph (DAG) constraint of NOTEARS (Zheng et al., 2018) with a scoring function. (5). **REASON** (Wang et al., 2023b): An interdependent network model learning both intra-level and inter-level causal relationships. (6). **Nezha** (Yu et al., 2023): A multi-modal method designed to identify root causes by detecting abnormal patterns. (7). **CORAL** (Wang et al., 2023a): An online single-modal RCA method based on incremental disentangled causal graph learning. (8). **CIRCA** (Li et al., 2022b): This model utilizes structural graph construction, regression-based hypothesis testing, and descendant adjustment to identify root cause metrics. (9). $\epsilon$**-Diagnosis** (Shan et al., 2019): This model diagnoses small-window, long-tail latency in large-scale microservice platforms using a two-sample test and $\epsilon$-statistics. (10). **RCD** (Ikram et al., 2022): This technique hierarchically localizes the root cause of failures by focusing on relevant sections of the causal graph. (11). **PCMCI** Runge et al. (2019): This technique combines conditional independence tests with a causal discovery algorithm to infer causal

networks from high-dimensional, nonlinear time series data. (12) **BARO** Pham et al. (2024): It is an end-to-end approach integrating Bayesian change point detection and nonparametric hypothesis testing to accurately detect anomalies and identify root causes in microservice systems.

The first four baseline models were originally designed to learn causal structures solely from time series data. As outlined in (Wang et al., 2023b;a), these causal discovery models can be extended to identify the root cause nodes. In this process, we first apply causal discovery models to learn the causal graphs, then utilize random walk with restarts (Wang et al., 2023a) on these graphs to identify the top $K$ nodes as root causes. The last three algorithms are applicable exclusively to metric data. Besides, we extend NOTEARS and GOLEM to the online learning setting, denoted by **NOTEARS**$^*$ and **GOLEM**$^*$, respectively[3]. For the online setting, we use the historical normal data (*e.g.*, 8 hours for the Product Review sub-dataset, and 1 hour for the SWaT and WADI sub-datasets) to construct the initial causal graph and update iteratively for each new batch of data. CORAL can inherit the causations from the previous data batch, while NOTEARS$^*$ and GOLEM$^*$ have to learn from scratch for each new data batch. More details of experimental settings can be found in Appendix E. For the hyperparameters, we use the default parameter values for all baselines to ensure a fair comparison.

Table 3: Results for offline RCA baselines with multiple modalities on the Product Review dataset.

| Modality | Model | PR@1 | PR@5 | PR@10 | MRR | MAP@3 | MAP@5 | MAP@10 |
|---|---|---|---|---|---|---|---|---|
| | Dynotears | 0 | 0 | 0.500 | 0.070 | 0 | 0 | 0.075 |
| | PC | 0 | 0 | 0.250 | 0.053 | 0 | 0 | 0.050 |
| | PCMCI | 0.250 | 0.500 | 0.500 | 0.342 | 0.250 | 0.300 | 0.400 |
| Metric Only | C-LSTM | 0.250 | 0.750 | 0.750 | 0.474 | 0.500 | 0.250 | 0.675 |
| | GOLEM | 0 | 0 | 0.250 | 0.043 | 0 | 0 | 0.025 |
| | RCD | 0 | 0 | 0.500 | 0.067 | 0 | 0 | 0.175 |
| | $\epsilon$-Diagnosis | 0 | 0 | 0 | 0.017 | 0 | 0 | 0 |
| | CIRCA | 0 | 0.500 | 0.500 | 0.250 | 0.333 | 0.400 | 0.450 |
| | BARO | 0.500 | 0.500 | 0.500 | 0.500 | 0.500 | 0.500 | 0.500 |
| | REASON | **0.750** | **1.000** | **1.000** | **0.875** | **0.917** | **0.950** | **0.975** |
| | Dynotears | 0 | 0 | 0.250 | 0.058 | 0 | 0 | 0.075 |
| | PC | 0 | 0 | 0.250 | 0.069 | 0 | 0 | 0.125 |
| Log Only | C-LSTM | 0 | 0 | 0.250 | 0.0590 | 0 | 0 | 0.075 |
| | GOLEM | 0 | 0 | 0.250 | 0.058 | 0 | 0 | 0.075 |
| | REASON | 0 | **0.500** | **0.750** | **0.216** | **0.167** | **0.250** | **0.400** |
| | Dynotears | 0 | 0 | 0.500 | 0.095 | 0 | 0 | 0.150 |
| | PC | 0 | 0 | 0.250 | 0.064 | 0 | 0 | 0.125 |
| | C-LSTM | 0.500 | 0.750 | 0.750 | 0.593 | 0.583 | 0.650 | 0.700 |
| Multi-Modality | GOLEM | 0 | 0 | 0.250 | 0.064 | 0 | 0 | 0.050 |
| | REASON | 0.750 | **1.000** | **1.000** | 0.875 | 0.917 | 0.950 | 0.975 |
| | Nezha | 0 | 0.500 | 0.750 | 0.193 | 0.083 | 0.250 | 0.475 |

## 4.2 OFFLINE ROOT CAUSE ANALYSIS RESULTS

**Product Review and Cloud Computing**. We evaluate nine offline RCA methods including both single-modal and multi-modal methods on Product Review and Cloud Computing sub-datasets. The experimental results are presented in Table 3 and Table 4 with respect to Precision at K (PR@K), Mean Reciprocal Rank (MRR), and Mean Average Precision at K (MAP@K). Our observations reveal the following insights: (1) PC algorithm and GOLEM have the worse performance on both Product Review and Cloud Computing sub-datasets. We conjecture that PC algorithm and GOLEM fail to capture the long term dependency for such a large-scale dataset, thus having difficulty of capturing the abnormal temporal patterns. Compared to PC algorithm and GOLEM, C-LSTM and Dynotears consider modeling the temporal dependency by their unique designs (i.e., Recurrent Structure for C-LSTM or dynamic Bayesian networks for Dynoters). Thus, we observe that C-LSTM and Dynotears outperform PC algorithm and GOLEM on both Product Review and Cloud Computing sub-datasets. This observation suggests the importance of modeling temporal dependency for these large-scale time-series datasets. (2) CIRCA outperforms RCD and $\varepsilon$-Diagnosis, which aligns with the results in the Petshop work (Saurabh Garg, Imaya Kumar Jagannathan, 2024), where CIRCA showed better accuracy in RCA due to its regression-based hypothesis testing and adjustment mechanisms. (3) The REASON method demonstrates notable success in identifying the root cause in 75% of system

---

[3] Other baselines are not extended to the online setting as they are time-intensive when there are multiple data batches.

fault scenarios on Product Review sub-dataset, achieving a PR@1 score of 75%. This indicates the utility of metric data alone in facilitating root cause identification. Compared to C-LSTM and Dynotears, we contribute the superiority of REASON to its design on multi-level causal structure learning. (3) The performance of these RCA methods is diminished when relying solely on log data for root cause analysis on both sub-datasets. This suggests that log data complements these methods, aiding in more accurate identification of potential root causes. (4) Integrating both metric and log data enhances the performance of most RCA methods in terms of MRR, compared to using only metric data. Additionally, we measure the difference between the dependency graph and the learned causal graph on the Product Review sub-dataset. The experimental results and discussion could be found in Appendix K.

Table 4: Results for offline RCA with multiple modalities on the Cloud Computing sub-dataset.

| Modality | Model | PR@1 | PR@5 | PR@10 | MRR | MAP@3 | MAP@5 | MAP@10 |
|---|---|---|---|---|---|---|---|---|
| | Dynotears | 0 | 0.167 | 0.333 | 0.075 | 0 | 0.033 | 0.117 |
| | PC | 0 | 0 | 0 | 0.029 | 0 | 0 | 0 |
| Metric Only | C-LSTM | 0.167 | 0.333 | 0.333 | 0.300 | 0.278 | 0.300 | 0.317 |
| | GOLEM | 0 | 0 | 0.167 | 0.044 | 0 | 0 | 0.017 |
| | RCD | 0 | 0 | 0 | 0.028 | 0 | 0 | 0 |
| | $\epsilon$-Diagnosis | 0 | 0 | 0 | 0.023 | 0 | 0 | 0 |
| | CIRCA | 0 | 0.167 | 0.333 | 0.090 | 0 | 0.033 | 0.167 |
| | REASON | 0.167 | **1.000** | **1.000** | 0.472 | 0.444 | 0.667 | 0.833 |
| | Dynotears | 0 | 0 | 0.167 | 0.048 | 0 | 0 | 0.050 |
| | PC | 0 | 0 | 0 | 0.032 | 0 | 0 | 0 |
| Log Only | C-LSTM | 0 | 0 | 0.167 | 0.044 | 0 | 0 | 0.050 |
| | GOLEM | 0 | 0 | 0.167 | 0.051 | 0 | 0 | 0.050 |
| | REASON | 0 | 0 | 0.333 | 0.082 | 0 | 0 | 0.067 |
| | Dynotears | 0 | 0.167 | 0.333 | 0.095 | 0 | 0.033 | 0.015 |
| | PC | 0 | 0 | 0.167 | 0.042 | 0 | 0 | 0.050 |
| | C-LSTM | 0.167 | 0.333 | 0.500 | 0.267 | 0.167 | 0.233 | 0.367 |
| Multi-Modality | GOLEM | 0 | 0 | 0.333 | 0.075 | 0 | 0 | 0.083 |
| | REASON | 0.333 | **1.000** | **1.000** | 0.597 | 0.611 | 0.767 | 0.883 |
| | Nezha | 0 | 0.333 | 0.333 | 0.148 | 0.111 | 0.020 | 0.267 |

Table 5: Results for offline RCA baselines on the SWaT sub-dataset.

| Dataset | Model | PR@1 | PR@5 | PR@10 | MRR | MAP@3 | MAP@5 | MAP@10 |
|---|---|---|---|---|---|---|---|---|
| | Dynotears | 0.125 | 0.323 | 0.427 | 0.279 | 0.201 | 0.244 | 0.308 |
| | PC | 0.125 | 0.344 | 0.583 | 0.262 | 0.129 | 0.204 | 0.350 |
| | C-LSTM | 0.125 | 0.281 | 0.521 | 0.294 | 0.139 | 0.177 | 0.319 |
| SWaT | GOLEM | 0.063 | 0.125 | 0.479 | 0.224 | 0.077 | 0.096 | 0.250 |
| | RCD | 0.125 | 0.125 | 0.625 | 0.228 | 0.125 | 0.125 | 0.344 |
| | $\varepsilon$-Diagnosis | 0.125 | 0.125 | 0.563 | 0.217 | 0.125 | 0.125 | 0.294 |
| | CIRCA | 0.188 | 0.250 | 0.688 | 0.287 | 0.188 | 0.200 | 0.394 |
| | REASON | **0.250** | **0.667** | **0.844** | **0.410** | **0.240** | **0.350** | **0.576** |

**Water Treatment/Distribution**. We employ eight single-modal RCA methods to assess root cause localization performance on the SWaT and WADI sub-datasets. The comparative results on the SWaT, presented in Table 5, are evaluated in terms of PR@K, MRR, and MAP@K. The experimental results on the WADI sub-datasets are presented in Table 7 in Appendix C. Consistent with observations on the Product Review and Cloud Computing sub-datasets, REASON outperforms the other four baseline methods. However, a decline in performance for the best baseline method, REASON, is noted when compared to its results on the Product Review and Cloud Computing datasets. This decrease in performance can be attributed to the nature of the SWaT and WADI sub-datasets, where faults are brief and the intervals between them are short. These fleeting events can be easily missed by most RCA methods, thus posing a significant challenge in accurately identifying the root causes within these two sub-datasets.

### 4.3 ONLINE ROOT CAUSE ANALYSIS RESULTS

We evaluate three RCA methods on all sub-datasets to demonstrate the utility of the LEMMA-RCA sub-dataset in an online setting. Notice that due to the lack of multi-modal online RCA methods,

we measure the performance of these single-modal baseline methods using only metric data shown in Table 6. By observation, we find that the online version of RCA models (*e.g.*, GOLEM*) outperform their offline version (*e.g.*, GOLEM) as online methods can rapidly capture the changing patterns of the metric data, thus learning a more accurate and noise-free causal structure for RCA. Among online methods, CORAL significantly outperforms NOTEARS* and GOLEM* due to the design of state-invariant and state-dependent representations learning tailored for the online setting. Notably, LEMMA-RCA is a large-scale real-world dataset, consisting of more than 100,000 timestamps across several days with various system fault scenarios, which can be naturally transformed to the online setting, compared with small datasets (*e.g.*, NeZha (Yu et al., 2023)) with limited timestamps for online RCA.

Table 6: Results for online root cause analysis baselines on all sub-datasets.

| Dataset | Model | PR@1 | PR@5 | PR@10 | MRR | MAP@3 | MAP@5 | MAP@10 |
|---|---|---|---|---|---|---|---|---|
| Product Review | CORAL | **0.750** | **1.000** | **1.000** | **0.875** | **0.917** | **0.950** | **0.975** |
| | NOTEARS* | 0.250 | 0.750 | 0.750 | 0.481 | 0.500 | 0.600 | 0.675 |
| | GOLEM* | 0.500 | 0.750 | 0.750 | 0.646 | 0.667 | 0.700 | 0.725 |
| Cloud Computing | CORAL | **0.500** | **0.833** | **1.000** | **0.667** | **0.667** | **0.733** | **0.867** |
| | NOTEARS* | 0 | 0.167 | 0.667 | 0.113 | 0 | 0.033 | 0.217 |
| | GOLEM* | 0 | 0.500 | 0.833 | 0.183 | 0.056 | 0.200 | 0.433 |
| SWaT | CORAL | **0.063** | **0.552** | **0.927** | **0.317** | **0.156** | **0.298** | **0.540** |
| | NOTEARS* | **0.063** | 0.365 | 0.677 | 0.263 | 0.149 | 0.235 | 0.422 |
| | GOLEM* | **0.063** | 0.427 | 0.688 | 0.281 | 0.170 | 0.260 | 0.437 |
| WADI | CORAL | **0.357** | **0.600** | **0.833** | **0.519** | **0.287** | **0.361** | **0.560** |
| | NOTEARS* | 0.143 | 0.457 | 0.726 | 0.377 | 0.187 | 0.275 | 0.484 |
| | GOLEM* | 0.241 | **0.600** | 0.738 | 0.402 | 0.198 | 0.303 | 0.490 |

## 5 DISCUSSIONS

**Broader impact**: To facilitate accurate, efficient, and multi-modal root cause analysis research across diverse domains, we introduce LEMMA-RCA as a new benchmark dataset. Our dataset also offers significant potential for advancing research in areas like **multi-modal anomaly detection**, **change point detection**, **causal structure learning**, and **LLM-based system diagnosis**. Based on the thorough data analysis and extensive experimental results, we highlight the following areas for future research:

- **Expanding Domain Applications**: To enhance the LEMMA-RCA dataset's versatility and impact, we plan to incorporate data from additional domains such as cybersecurity and healthcare. This integration of diverse data sources will facilitate the development of more comprehensive root cause analysis technologies, significantly extending the dataset's applicability across various industries.

- **Online Multi-Modal Root Cause Analysis**: Most RCA methods are offline and single-modal, leaving a gap for real-time, multi-modal approaches. Developing these methods can enable instant analysis of diverse data streams, essential for dynamic environments like industrial automation and real-time monitoring.

**Limitations**: Despite its broad capabilities, the LEMMA-RCA dataset may have limited generalizability, as its fault scenarios may not fully capture the diversity of real-world conditions due to factors like system interruptions and unforeseen circumstances. Additionally, the dependency graphs in our data are semi-complete, reflecting the inherent challenge of obtaining complete ground-truth graphs in complex systems, which may impact the precision of derived analyses.

## 6 CONCLUSION

In this work, we present LEMMA-RCA, the first large-scale, open-source dataset featuring real system faults across various application domains and multiple modalities. We conduct an inclusive empirical study on LEMMA-RCA by testing the performance of fourteen baseline methodologies under different settings, including offline/online modes and single/multiple-modality data. Our experimental results demonstrate the utility of LEMMA-RCA. By making this dataset publicly available, we aim to facilitate further research and innovation in root cause analysis for complex systems, contributing significantly to the development of more robust and secure methodologies that ensure the high performance of modern systems, particularly those that are mission-critical.

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

## A    MONITORING TIME SERIES SEGMENTATION FOR SWaT AND WADI

In the original SWaT and WADI datasets, the attack model demonstrates irregular attack patterns, occasionally targeting multiple sensors simultaneously, or executing attacks at closely spaced intervals. To follow the principles of RCA, we have established two specific preprocessing rules for these datasets: 1) Each recorded attack event must only involve a single sensor or actuator. 2) The duration of the dataset corresponding to each attack event must be standardized to two hours. Consequently, we selectively keep attack events that impact only one sensor or actuator. If the interval between successive attack events is insufficiently short, we assume the stability in the monitoring data immediately before and after each attack event. To ensure the necessary two-hour duration for each event, we concatenate normal-state data from both before and after the attack period. This adjustment positions the attack event centrally within a continuous two-hour segment, facilitating consistent and accurate analysis.

## B    ADDITIONAL SYSTEM FAULT SCENARIOS

This section describes the processes used to generate and monitor system fault scenarios, with emphasis on mimicking real-world fault patterns. Each scenario involved the induction of specific failure conditions, while allowing the microservice system to exhibit its natural behavior under stress. Metrics and logs were collected using established monitoring tools, such as Prometheus, Elasticsearch, CloudWatch, Jaeger, and JMeter.

- **Silent Pod Degradation Fault.**
    - **Description:** A pod in a load balancer contains a latent bug causing its CPU usage to rise, which gradually increases latency for a subset of users without triggering autoscaling or error alerts.
    - **Method:** We periodically sent requests to Microservice A over a 24-hour period. After this initial observation, we manually increased the CPU load on one specific `productpage-v1` pod to simulate the bug.
    - **Data Collection:** Metrics and logs were collected from CloudWatch, while KPIs such as latency were measured using JMeter. The goal was to trace latency increases back to the specific pod with elevated CPU utilization.

- **Noisy Neighbor Issue.**
    - **Description:** A neighboring pod in a shared node generates high CPU load, impacting the performance of the `productpage-v1` pod and causing elevated error rates.
    - **Method:** Requests were sent to Microservice A, while the pod ratings of Microservice B (`robot-shop`) were moved to the same node as `productpage-v1`, generating contention.
    - **Data Collection:** Metrics (CPU usage, memory usage) were gathered using Prometheus, while logs were obtained from CloudWatch Logs. Configuration changes, such as node assignments, were also recorded.

- **Node Resource Contention Stress Test.**
    - **Description:** A stress test on CPU resources was conducted by inducing high load on Microservice B, co-located with Microservice A on the same node.
    - **Method:** Periodic requests were sent to Microservice A using JMeter, while a high CPU load was generated on Microservice B using the `OpenSSL speed` command.
    - **Data Collection:** HTTP response logs from JMeter were analyzed for performance impacts. System metrics (CPU and memory usage) were retrieved from Prometheus, while container logs were collected from Elasticsearch.

- **DDoS Attack.**
    - **Description:** A Distributed Denial of Service (DDoS) attack was simulated to overload the system, causing Out-of-Memory (OOM) errors in targeted pods.
    - **Method:** Over a monitoring period of approximately 48 hours, we gradually increased the request rate to Microservice A, eventually overwhelming the `reviews-v2` and `reviews-v3` pods.

- **Data Collection:** Metrics such as CPU and memory utilization were collected via Prometheus. Logs from Jaeger and Elasticsearch provided insights into the system's response to the attack.

- **Malware Attack.**

  - **Description:** A malware pod executed a password list attack to compromise other pods, propagating DDoS scripts to degrade overall system performance.
  - **Method:** The attack started from a designated pod (`scenario10-malware-deployment`) and targeted others via SSH password brute-forcing, ultimately generating high load on `productpage-v1`.
  - **Data Collection:** JMeter was used to monitor KPIs (latency, error rate), while Prometheus and CloudWatch Logs provided system metrics and logs for root-cause analysis.

- **Bug Infection.**

  - **Description:** A latent bug in the API caused asymmetric CPU load increases, degrading response times without fully utilizing the CPU capacity.
  - **Method:** Requests were sent periodically to the web service, and after a day, a script induced increased CPU utilization on one core.
  - **Data Collection:** KPIs were measured using JMeter, while system metrics and logs were collected via CloudWatch for detailed analysis.

- **Configuration Fault.**

  - **Description:** An incorrect resource limit in a Kubernetes manifest file led to a pod being terminated by the OOM killer, impacting other services.
  - **Method:** Requests were sent to Microservice A, while a Git push introduced a faulty configuration for the `details-v1` pod. The misconfigured pod eventually failed under load.
  - **Data Collection:** Error rates were tracked using JMeter, and metrics/logs were retrieved from Prometheus and CloudWatch for root-cause identification.

## C  ADDITIONAL EXPERIMENTAL RESULTS

Here, we provide the additional experimental results of offline RCA methods on the WADI dataset in Table 7.

Table 7: Results for offline root cause analysis baselines on the WADI sub-dataset.

| Dataset | Model | PR@1 | PR@5 | PR@10 | MRR | MAP@3 | MAP@5 | MAP@10 |
|---------|-------|------|------|-------|-----|-------|-------|--------|
| WADI | Dynotears | 0.071 | 0.300 | 0.476 | 0.222 | 0.107 | 0.174 | 0.268 |
| | PC | 0.071 | 0.350 | 0.500 | 0.277 | 0.163 | 0.239 | 0.346 |
| | C-LSTM | 0 | 0.350 | 0.512 | 0.244 | 0.115 | 0.186 | 0.327 |
| | GOLEM | 0 | 0.400 | 0.536 | 0.235 | 0.099 | 0.204 | 0.348 |
| | RCD | 0.071 | 0.400 | 0.643 | 0.264 | 0.190 | 0.286 | 0.464 |
| | $\epsilon$-Diagnosis | 0 | 0.350 | 0.500 | 0.211 | 0.167 | 0.249 | 0.371 |
| | CIRCA | 0.143 | 0.550 | 0.714 | 0.350 | 0.301 | 0.400 | 0.529 |
| | REASON | **0.286** | **0.650** | **0.798** | **0.534** | **0.425** | **0.506** | **0.638** |

## D  LEMMA-RCA LICENSE

The LEMMA-RCA benchmark dataset is released under a CC BY-ND 4.0 International License: https://creativecommons.org/licenses/by-nd/4.0. The license of any specific baseline methods used in our codebase should be verified on their official repositories.

## E   REPRODUCIBILITY

All experiments are conducted on a server running Ubuntu 18 with an Intel(R) Xeon(R) Silver 4110 CPU @2.10GHz and one 11GB GTX2080 GPU. In the online RCA experiment, we set the size of historical metric and log data to 8-hour intervals and each batch is set to be a 10-minute interval. We use the Adam as the optimizer and we train the model for 100 iterations at each batch. In addition, all methods were implemented using Python 3.8.12 and PyTorch 1.7.1.

## F   DETAILED DESCRIPTION OF BASELINES

We evaluate the performance of the following RCA models on the benchmark sub-datasets:

- **PC** (Burr, 2003): The PC algorithm is a data-driven method for causal discovery, producing a partially directed acyclic graph (PDAG) that represents causal relationships among variables. It starts with a fully connected graph and iteratively removes edges based on conditional independence tests, then orients the remaining edges to construct a causal structure. The algorithm assumes the causal Markov property, no hidden confounders, and no cycles in the graph. It is widely used for root cause analysis to identify direct and indirect influences on specific outcomes but is sensitive to the reliability of independence tests and cannot distinguish between equivalent causal structures.

- **Dynotears** (Pamfil et al., 2020): Dynotears is score-based approach for learning these models that scales gracefully to high-dimensional datasets. To accomplish this, the authors cast the problem as an optimization problem (i.e. score-based learning), and use standard second-order optimization schemes to solve the resulting program. Dynotears is based on the recent algebraic characterization of acyclicity in directed graphs, which makes the formulation simple and amenable to different modeling choices.

- **C-LSTM** (Tank et al., 2022): The C-LSTM framework is designed for interpretable nonlinear Granger causality discovery in MLPs and RNNs by leveraging the flexibility of neural networks while introducing component-wise architectures to disentangle the effects of lagged inputs on individual outputs. It enhances interpretability and manages limited, high-dimensional data by applying sparsity-inducing penalties to weight groupings that connect input histories to output series. The framework's sparse component-wise models, such as cMLP and cLSTM, incorporate group sparsity penalties to effectively select Granger-causal relationships through the outgoing weights of inputs.

- **GOLEM** (Ng et al., 2020): GOLEM (Gradient-based Optimization of dag-penalized Likelihood for learning linEar dag Models) is a likelihood-based structure learning method for DAGs that replaces hard DAG constraints with soft sparsity and DAG penalties, enabling continuous unconstrained optimization. This approach simplifies the optimization problem while maintaining the ability to learn a DAG equivalent to the ground truth. The framework is validated in both asymptotic and finite-sample regimes, demonstrating its flexibility across various linear models. By avoiding strict constraints, GOLEM is computationally more efficient and theoretically robust for causal discovery.

- **REASON** (Wang et al., 2023b): REASON is a framework for root cause localization in complex systems with interdependent network structures. It combines Topological Causal Discovery (TCD) and Individual Causal Discovery (ICD). TCD employs hierarchical graph neural networks to uncover intra- and inter-level causal relationships, modeling fault propagation using a random walk with restarts. ICD focuses on analyzing individual time-series data, using Extreme Value theory to detect abrupt fluctuations and estimate root cause likelihoods, especially for short-lived failures. The framework integrates results from both components to identify system entities with the highest causal scores as root causes.

- **Nezha** (Yu et al., 2023): Nezha is an interpretable and fine-grained root cause analysis (RCA) method for microservices that unifies heterogeneous observability data (metrics, traces, logs) into a homogeneous event format. This representation enables the construction of event graphs for integrated analysis. Nezha statistically localizes actionable root causes at granular levels, such as specific code regions or resource types, offering high interpretability to support confident mitigation actions by SREs.

- **CORAL** (Wang et al., 2023a): CORAL is an online root cause analysis (RCA) framework that automatically triggers RCA processes and incrementally updates the RCA model. It includes three key components: Trigger Point Detection, Incremental Disentangled Causal Graph Learning, and Network Propagation-based Root Cause Localization. The trigger detection uses multivariate singular spectrum analysis and cumulative sum statistics to identify system state transitions in near-real-time. Incremental causal graph learning decouples state-invariant and state-dependent information to efficiently update the RCA model. Finally, CORAL applies a random walk with restarts on the causal graph to localize root causes, terminating when the causal graph and root cause list stabilize.

- **CIRCA** (Li et al., 2022b): CIRCA is an unsupervised root cause analysis method that formulates the problem as a causal inference task called intervention recognition. Its core idea is to identify root cause indicators by evaluating changes in the probability distribution of monitoring variables conditioned on their parents in a Causal Bayesian Network (CBN). CIRCA applies this approach to online service systems by constructing a graph among monitoring metrics, leveraging system architecture knowledge and causal assumptions to guide the analysis.

- $\epsilon$-**Diagnosis** (Shan et al., 2019): $\epsilon$-Diagnosis is an unsupervised, low-cost diagnosis algorithm designed to address small-window long-tail latency (SWLT) in web services, which arises in short statistical windows and typically affects a small subset of containers in microservice clusters. It uses a two-sample test algorithm and $\epsilon$-statistics to measure the similarity of time series, enabling the identification of root-cause metrics from millions of metrics. The algorithm is implemented in a real-time diagnosis system for production microservice platforms.

- **RCD** (Ikram et al., 2022): RCD is a scalable algorithm for detecting root causes of failures in complex microservice architectures using a hierarchical and localized learning approach. It treats the failure as an intervention to quickly identify the root cause, focuses learning on the relevant portion of the causal graph to avoid costly conditional independence tests, and explores the graph hierarchically. The technique is highly scalable, providing actionable insights about root causes, while traditional methods become infeasible due to high computation time.

## G    FIGURES FOR CLARITY

We provide figures related to the system architecture and fault scenarios in this section, for better readability. The architecture of Product Review Platform is shown in Figure 6, and the system fault scenarios are demonstrated in Figure 7 and Figure 8.

## H    DATASET LABELING METHODOLOGY

We provide more details on the system fault labeling strategy, which comes in two-fold: the root cause labeling process and label validation.

**Root Cause Labeling Process.**

- For each system fault, we designed controlled fault scenarios to mimic realistic fault patterns (e.g., external storage failure, database overload).

- During each controlled fault case, we monitored system behaviors, including metrics and logs, to identify the exact root cause of the fault.

- The ground truth root cause was then labeled based on the specific fault of the system. This ensures high accuracy in root cause labeling, as the faults were systematically induced and their impacts directly observed.

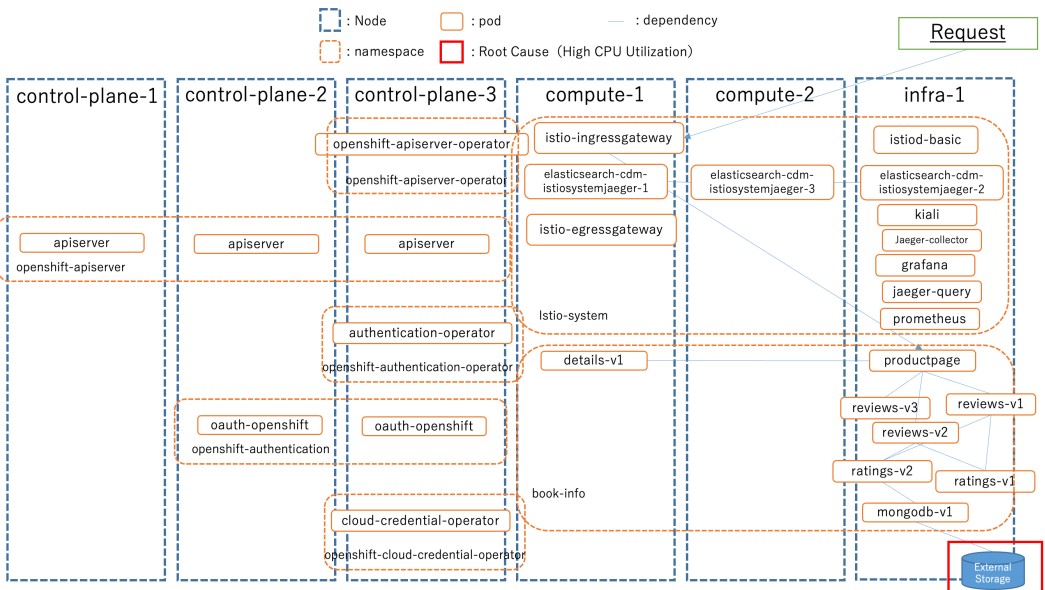

Figure 6: Corresponding to Figure 2 (a). The architecture of Product Review Platform

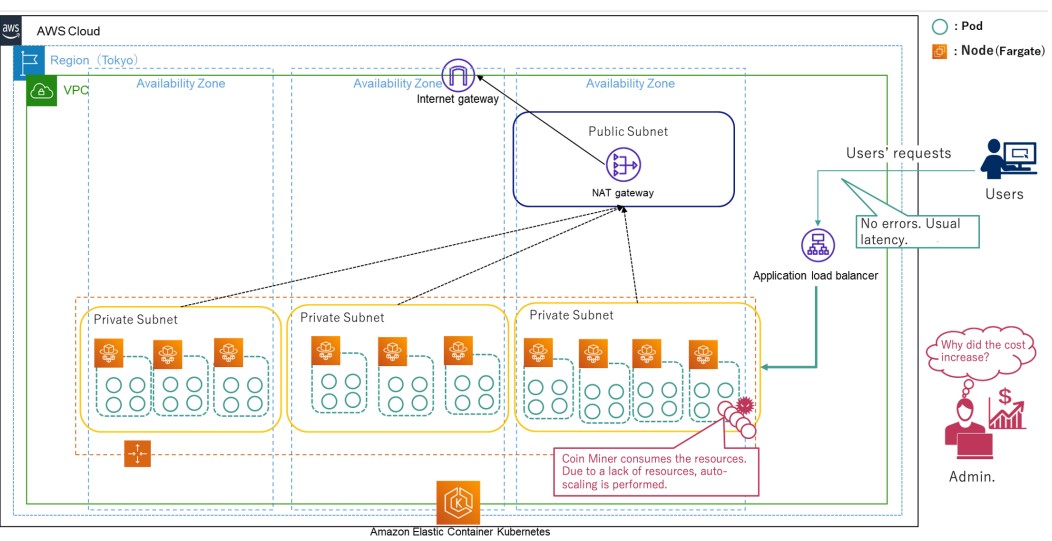

Figure 7: Corresponding to Figure 4 left. Visualization of Cryptojacking system fault scenario. **Right:** External storage failure.

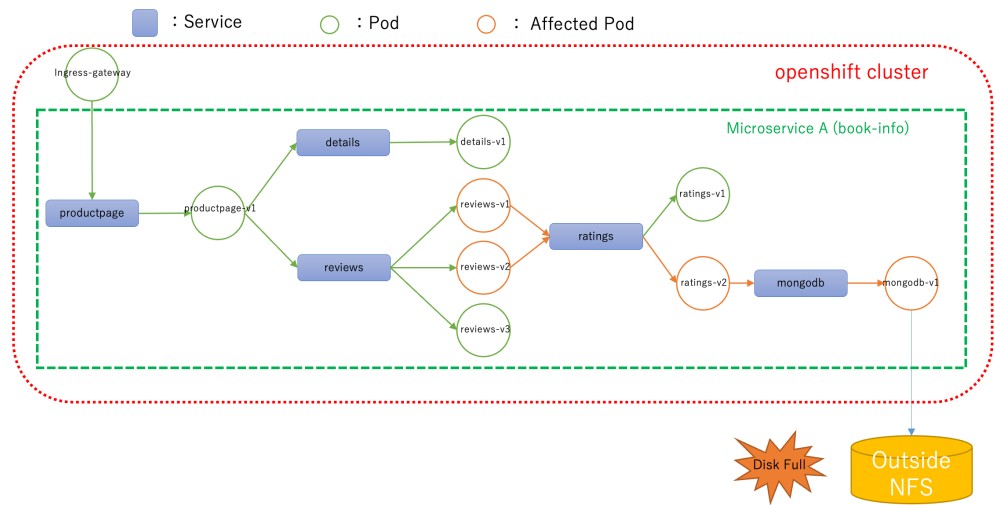

Figure 8: Corresponding to Figure 4 right. Visualization of External storage failure. system fault scenario.

**Label Validation.**

- To ensure label correctness, we validated the root cause labels by analyzing the system's behavior during and after fault. This involved cross-checking the observed anomalies in system metrics and logs with the expected outcomes of the fault.
- Multiple experts reviewed the labeled faults to confirm the consistency and correctness of the root cause assignments.

## I PARAMETER SETTINGS FOR ALL BASELINE MODELS

We provide the detailed parameter settings for all baseline models as follows:

- **Dynotears:** `lag=20` (*maximum time lags*), `lambda_w=1e-3` (*weight regularization*), `lambda_a=1e-3` (*autoregressive term regularization*), `g_thre=0.3` (*sparsity threshold*).
- **PC:** `alpha=0.05` (*significance level for conditional independence tests*), `ci_test='fisherz'` (*type of conditional independence test*).
- **C-LSTM:** `hidden=100` (*hidden units in LSTM*), `lag=20` (*maximum time lags for sequence modeling*), `lam=10.0` (*model complexity regularization*), `lam_ridge=1e-2` (*ridge regression regularization*), `lr=1e-3` (*learning rate*), `max_iter=30000` (*maximum iterations*), `g_thre=0.3` (*sparsity threshold*).
- **GOLEM:** `lambda_1=2e-2` (*weight for sparsity regularization*), `lambda_2=5.0` (*weight for smoothness regularization*), `learning_rate=1e-3` (*optimization learning rate*), `num_iter=30000` (*number of iterations for training*), `g_thre=0.3` (*sparsity threshold*).
- **REASON:** `lag=20` (*maximum time lags for causal modeling*), `L=150` (*hidden layers with 150 units*), `lambda1=1` (*adjacency matrix sparsity regularization*), `lambda2=1e-2` (*autoregressive term balancing regularization*), `gamma=0.8` (*integration of individual and topological causal effects*), `g_thre=0.3` (*sparsity threshold*).

## J PARAMETER SENSITIVITY ANALYSIS ON PRODUCT REVIEW SUBDATASET (USING REASON)

We conducted parameter sensitivity tests for $\gamma$ and $L$ on the Product Review subdataset. The results are summarized in the following tables:

$\gamma$ SENSITIVITY

| $\gamma$ | MAP@10 | MRR |
|---|---|---|
| 0.1 | 0.80 | 0.81 |
| 0.2 | 0.80 | 0.81 |
| 0.3 | 0.84 | 0.82 |
| 0.4 | 0.86 | 0.83 |
| 0.5 | 0.88 | 0.83 |
| 0.6 | 0.88 | 0.73 |
| 0.7 | 0.86 | 0.83 |
| 0.8 | 0.92 | 0.84 |
| 0.9 | 0.90 | 0.74 |

Table 8: Sensitivity of $\gamma$ on Product Review subdataset.

**Analysis:** The optimal $\gamma$ value is 0.8, achieving the best MAP@10 (0.92) and MRR (0.84). This result demonstrates that a balanced integration of individual and topological causal effects is critical for performance.

$L$ SENSITIVITY

| $L$ | MAP@10 | MRR |
|---|---|---|
| 10 | 0.52 | 0.50 |
| 20 | 0.33 | 0.25 |
| 50 | 0.37 | 0.32 |
| 100 | 0.42 | 0.28 |
| 150 | 0.53 | 0.50 |
| 200 | 0.37 | 0.33 |

Table 9: Sensitivity of $L$ on Product Review subdataset.

**Analysis:** The best performance is observed at $L = 150$, where MAP@10 and MRR reach 0.53 and 0.50, respectively. This indicates that $L = 150$ provides the optimal hidden layer size, balancing model capacity and complexity while avoiding underfitting or overfitting.

# K QUALITY EVALUATION BASED ON THE COMPARISON BETWEEN DEPENDENCY GRAPH AND LEARNED CAUSAL GRAPH

To evaluate the difference between the semi-complete dependency graph and the causal graph learned by baseline methods, we conducted experiments on the Product Review sub-dataset (system metrics data only). Following the methodology outlined in [1], we assessed the performance using four commonly used metrics: True Positive Rate (TPR), False Discovery Rate (FDR), Structural Hamming Distance (SHD), and Area Under the ROC Curve (AUROC).

| Method | TPR ↑ | FDR ↓ | SHD↓ | AUROC ↑ |
|---|---|---|---|---|
| Dynotear | 0.214 | 0.743 | 0.786 | 0.612 |
| PC | 0.112 | 0.892 | 0.861 | 0.563 |
| C-LSTM | 0.428 | 0.427 | 0.543 | 0.733 |
| GOLEM | 0.126 | 0.847 | 0.823 | 0.571 |
| RCD | 0.152 | 0.869 | 0.838 | 0.584 |
| $\epsilon$-Diagnosis | 0.084 | 0.905 | 0.874 | 0.554 |
| CIRCA | 0.327 | 0.544 | 0.582 | 0.685 |
| REASON | 0.634 | 0.217 | 0.347 | 0.846 |

Table 10: Comparison Between Dependency Graph and Learned Causal Graph on the Product Review sub-dataset.

**Evaluation and Results:** For each system fault, we computed the metrics individually and then averaged the results across four cases. It is important to note that system entities not included in the semi-complete dependency graph were excluded from this comparison to ensure consistency and fairness across methods. To ensure comparability for SHD, which is influenced by the number of nodes in the graph, we normalized SHD by dividing it by the square of the number of nodes for each system fault. Finally, we averaged the normalized SHD across the four system faults on the Product Review sub-dataset. These results are summarized in the table above, providing a comprehensive comparison between the dependency and causal graphs.

## L    DATASET REPRESENTATIVENESS

In this section, we aim to show the representativeness of the released dataset. While it is challenging to establish a universal metric for representativeness in benchmarks, we have made significant efforts to ensure the dataset covers diverse fault scenarios:

- **Real-World Fault Scenarios:** The IT domain datasets (Product Review and Cloud Computing) encompass realistic microservice faults such as out-of-memory errors, DDoS attacks, and cryptojacking, as outlined in Section 3.1 and Appendix B. Similarly, the OT domain datasets (SWaT and WADI) include real-world cyber-physical system faults recorded in controlled environments.

- **Diversity of Fault Types:** Across IT and OT domains, we include 10 distinct fault types, ensuring coverage of both transient and persistent system failures. This diversity reflects common issues faced by modern IT and OT systems.

- **Comparative Analysis:** As seen in Table 3 and related discussions, our dataset exhibits performance trends consistent with other benchmarks (e.g., Petshop), supporting its credibility as a representative evaluation platform.

- **Quality Assurance:** All data were collected using industry-standard monitoring tools like Prometheus, CloudWatch, and Elasticsearch. Each fault scenario was validated to ensure it mirrors real-world conditions.

