# OpenReview forum: "LEMMA-RCA: A Large Multi-modal Multi-domain Dataset for Root Cause Analysis"
_ICLR.cc/2025/Conference — Submitted to ICLR 2025_

### Official Review · Reviewer_BP9D · 2024-10-26

**Soundness:** 2
**Presentation:** 2
**Contribution:** 3
**Rating:** 5
**Confidence:** 3

**Summary:**

The paper presents Lemma-RCA, a novel dataset and benchmark designed for root cause analysis (RCA). This dataset includes four sub-datasets: two from IT environments and two from OT environments, offering a large-scale, multi-modality dataset (with both KPI and log data) that captures real-world system faults. The authors validate the dataset’s quality by testing it with eight baseline models across offline single-/multi-modality and online single-modality settings.

**Strengths:**

- The dataset is open-source, multi-modal, and well-suited to RCA, making it both timely and relevant.
- The authors have provided a thorough review of existing baseline approaches.

**Weaknesses:**

- The dataset description could be more accessible to a broader audience, as suggested in the questions below.
- Reproducibility is limited due to insufficient implementation details for the baseline models.

**Questions:**

1. **Clarification:**
- The current data collection section seems tailored for domain experts and could benefit from clarification for a general audience. For instance, what do "IT" and "OT" refer to? Are "Prometheus" and "ElasticSearch" tools or companies? Clarifying the meanings of such terms would improve the accessibility.
- Figure 2(a) would benefit from a more detailed explanation in its caption.
- In Figure 3, what is the KPI being referenced? Should it be assumed that all KPIs in this figure relate to system latency? Please specify the y-axis further.
- How were the root causes $V_a$  for each system fault $a$ labeled? The authors  may include a section in the paper detailing their labeling methodology.
2. **Evaluation Metrics:** The evaluation metrics appear to be sample-independent. Why did the authors not consider sample-dependent metrics? For example, over a 10-day system run yielding 1000 faults, the accuracy of the prediction algorithm could be tested against the actual root cause labels.

3. **Data Quality Claims:** The authors suggest high data quality based on baseline comparisons. This conclusion seems somewhat overstated, as the main insight from these experiments appears to be that "MRR performance improves when considering two modalities jointly."

**Comment on Missing Data:** While the authors view missing data as a limitation, I consider it a realistic aspect of real-world data, which poses a meaningful challenge rather than a flaw.

---

> ### Author Response · Authors · 2024-11-19
> **Rebuttal by Authors**
>
> Dear reviewer BP9D:
>
> Thank you for your invaluable feedback. We would like to address your primary concerns and provide corresponding responses below.
> \
> \
> **Response to: What do "IT" and "OT" refer to? Are "Prometheus" and "ElasticSearch" tools or companies? Figure 2(a) would benefit from a more detailed explanation in its caption. In Figure 3, what is the KPI being referenced? How were the root causes for each system fault labeled? The authors may detail their labeling methodology.**
> \
> \
> A: Thank you for your detailed suggestions regarding terminology clarification, system KPI definitions, figure descriptions, and the labeling methodology. We appreciate the opportunity to improve the accessibility and clarity of our work. Below, we outline the updates made to address your feedback:
>
> 1. Terminologies and System KPI Clarifications:
> - IT and OT:
>   - IT (Information Technology) refers to technology systems used for managing data, computing, and communications in an organizational context.
>   - OT (Operational Technology) refers to systems that control physical devices, processes, and infrastructure in industries such as manufacturing, utilities, and transportation.
>   - We have clarified these terms in the abstract and the main text where they first appear.
> - Prometheus and ElasticSearch:
>   - Prometheus [1] is an open-source, metrics-based monitoring service commonly used in cloud platforms to track system performance.
>   - ElasticSearch [2] is an open-source big data processing tool designed for text search and analysis. Its efficiency in managing log data has led to its widespread use in system monitoring [3].
>   - These explanations have been added to the revised draft.
> - System KPI and Figures:
>   - The system KPI referenced in Figure 3 represents **system latency** (response time), measured in **seconds**.
>   - The y-axis in Figure 3 has been clarified to explicitly indicate it is the system KPI (latency).
>
> 2. Root Cause Labeling Methodology:
>
> To address your question on labeling, we have added detailed descriptions of the **labeling process** and **validation** to the revised manuscript (Appendix H).
>
> - Root Cause Labeling Process:
>   - For each system fault, controlled fault scenarios were designed to mimic realistic fault patterns, such as external storage failure or database overload.
>   - During these fault scenarios, system behaviors—including metrics and logs—were monitored to identify the exact root cause of the fault.
>   - The **ground truth root cause** was labeled based on the specific fault induced, ensuring high accuracy through direct observation of the fault’s impact.
> - Label Validation:
>   - To validate the labels, system behaviors during and after the fault were analyzed. Observed anomalies in metrics and logs were cross-checked against the expected outcomes of the fault scenario.
>   - Multiple domain experts reviewed the labeled faults to confirm their correctness and consistency.
>
> 3. Figure Improvements:
>
> - Figure 2(a): The caption has been expanded to include a more detailed explanation of the figure, including the context and relevance of the data presented.
> - Figure 3: The y-axis has been clarified as representing system latency (seconds), and the KPI reference has been explicitly tied to this metric.
>
> We hope these updates address your concerns. Please let us know if there are additional clarifications or enhancements you would like to see.
>
> [1] https://www.prometheusbook.com/
>
> [2] https://www.elastic.co/elasticsearch
>
> [3] Vlad-Andrei Zamfir, Mihai Carabas, Costin Carabas, and Nicolae Tapus. Systems monitoring and big data analysis using the elasticsearch system. In 2019 22nd International Conference on Control Systems and Computer Science (CSCS), pp. 188–193. IEEE, 2019.
> \
> \
> **Response to: While the authors view missing data as a limitation, I consider it a realistic aspect of real-world data, which poses a meaningful challenge rather than a flaw.**
> \
> \
> A: Thank you for your insightful comment regarding missing data. We agree that missing data is a realistic aspect of real-world systems and can present meaningful challenges for root cause analysis, rather than being viewed as a flaw.
>
> In light of your feedback, we have updated the manuscript to reflect this perspective. Specifically:
> - We have removed the mention of missing data as a limitation in the updated version.
> - We have reframed missing data as an inherent characteristic of real-world datasets, highlighting its potential to challenge and improve the robustness of RCA methods.
>
> We appreciate your perspective and believe it adds value to the interpretation of our dataset's characteristics. Please let us know if there are further improvements you would like to see.

---

> > ### Author Response · Authors · 2024-11-19
> > **Rebuttal by Authors**
> >
> > **Response to: The evaluation metrics appear to be sample-independent. Why did the authors not consider sample-dependent metrics?**
> > \
> > \
> > A: Thank you for your suggestion regarding the use of sample-dependent metrics. We would like to clarify why we chose case-based evaluation metrics and why sample-dependent metrics may not align with real-world RCA scenarios.
> >
> > 1. RCA as a Diagnosis/Post-Analysis Task:
> > - Root Cause Analysis (RCA) is fundamentally a **diagnosis or post-analysis task**, rather than a detection task.
> > - Unlike detection tasks, which involve monitoring and identifying anomalies across many samples, RCA focuses on **explaining and diagnosing the root cause of a specific fault** after it has occurred.
> > - This diagnostic nature inherently makes RCA a case-by-case process, where each fault is treated as an independent case requiring detailed analysis to identify its root cause.
> > 2. Real-World Applicability of Fault Frequency:
> > - In real-world systems, it is uncommon to encounter **1000 faults over a 10-day period**, as suggested. Such a high fault frequency is unrealistic for many production systems, where faults are typically infrequent but have significant impacts when they occur.
> > - Designing evaluation metrics based on an assumption of frequent faults may not accurately reflect the operational context of RCA tasks.
> > 3. Dependent Nature of Malfunction Events in a Fault Scenario:
> > - While RCA is performed on a case-by-case basis, the malfunction events within a specific fault are often dependent. The root cause is typically defined as the first step or primary trigger in a sequence of events leading to the fault.
> > - For example, in a DDoS attack scenario, the initial step of the DDoS attack (e.g., a sudden surge in malicious traffic) is considered the root cause, even though it leads to subsequent system failures like server overload.
> > - Case-based evaluation metrics reflect this dependency structure by focusing on identifying the root cause as the initiating event in each fault scenario.
> > 4. Independence of Repeated Testing:
> > - Testing the same fault case multiple times (e.g., 100 times) does not provide additional insights into RCA performance, as the results for a specific case remain constant.
> > - The focus in RCA is the **correctness of the root cause identification** for each unique fault case, not the frequency of testing or fault occurrences.
> >
> > We hope this clarifies our reasoning for selecting case-based evaluation metrics and their alignment with the real-world goals of RCA. If additional points need further elaboration, we are happy to address them.
> > \
> > \
> > **Response to: The authors suggest high data quality based on baseline comparisons. This conclusion seems somewhat overstated, as the main insight from these experiments appears to be that "MRR performance improves when considering two modalities jointly."**
> > \
> > \
> > A: Thank you for your feedback regarding the interpretation of our experimental results and data quality claims.
> > We agree that the statement about data quality based on baseline comparisons could be better clarified. To address this:
> >
> > 1. Clarification of Data Quality Claims:
> > - Our data quality claims are not solely based on the baseline comparisons but also on the design of the dataset, which includes diverse and realistic fault scenarios collected from real-world systems.
> > - The experiments demonstrate that the joint use of metrics and log modalities improves performance (e.g., MRR), highlighting the complementary nature of these data modalities and the richness of the dataset in capturing fault-relevant patterns.
> > 2. Additional Explanation in Section 4.2:
> > - We have expanded the explanation of the experimental results in the updated version of Section 4.2 (highlighted in red). This includes a more detailed discussion of how the dataset facilitates performance differentiation across various methods and modalities.
> > 3. Data Quality Beyond Metrics:
> > - Beyond MRR improvements, the dataset’s quality is evident in its scale, diversity, and real-world relevance, which are not fully captured by any single performance metric. For example, the dataset includes high temporal resolution and a large number of nodes, providing comprehensive coverage of system behaviors and fault scenarios.
> >
> > We hope these clarifications address your concerns and provide a more nuanced understanding of our data quality claims. If additional explanations are needed, we are happy to elaborate further.

---

> > > ### Author Response · Authors · 2024-11-19
> > > **Rebuttal by Authors**
> > >
> > > **Response to: Reproducibility is limited due to insufficient implementation details for the baseline models.**
> > > \
> > > \
> > > A: Thank you for your feedback regarding the reproducibility of our baseline model implementations.
> > >
> > > We would like to clarify that we have released the source code for all the baseline methods used in our experiments to ensure reproducibility. The source code is accessible via the anonymous GitHub link provided in the abstract. To access it:
> > >
> > > - Open the link provided in the abstract.
> > > - Click on the Source option in the menu at the top-right corner.
> > > - From there, click on GitHub to access the repository containing the implementation details.
> > >
> > > This ensures that all baseline implementations are transparent and reproducible. If there are specific aspects of the implementation that require further clarification, please let us know, and we will be happy to address them.

---

> ### Comment · Reviewer_BP9D · 2024-11-19
>
> Thank you to the authors for their response and for addressing my questions. I appreciate the efforts made. Here are my revised comments:
>
> 1. **Issue of Overstatement:** The statement “We evaluate the quality of LEMMA-RCA by testing the performance of eight baselines” seems to overstate the implications of the baseline tests. While I acknowledge the dataset’s quality through expert inspection and validation, I don’t think the baseline comparisons alone can determine the relative quality of datasets. I recommend revising similar statements.
> 2. **Baseline Implementation Details:** I suggest including the key parameters used for the baselines in the appendix, as these are crucial for replicating the results and understanding the methods. It would be informative to know if parameter tuning was performed for each baseline. Additionally, incorporating a discussion on the sensitivity to hyper-parameter tuning in a dedicated column could enhance the clarity and completeness of the comparison.

---

> > ### Author Response · Authors · 2024-11-20
> > **Response to Reviewer BP9D**
> >
> > **Response to: Issue of Overstatement**
> > \
> > \
> > Thank you for your prompt and insightful feedback on avoiding overstatement regarding the evaluation of LEMMA-RCA. We agree that the performance of baselines alone cannot fully determine the quality of a dataset, and we appreciate your emphasis on the importance of precise wording. To address your concern, we have revised the original statement to ensure it accurately reflects the scope of our empirical study without implying that baseline performance serves as a direct evaluation of dataset quality.
> >
> > The revised statement now reads:
> > >“We evaluate the performance of ten baseline methods on LEMMA-RCA.”
> >
> > This revision removes any implication that the baseline tests are intended to evaluate the dataset’s quality and instead presents them as a component of our empirical study.
> >
> > We hope this change aligns with your recommendation and adequately addresses your concern. Please let us know if further clarification or revisions are needed.
> > \
> > \
> > **Response to: Baseline Implementation Details**
> > \
> > \
> > Thank you for your valuable feedback on including parameter details and discussing sensitivity to hyper-parameter tuning. We have made the following updates to address your concerns:
> >
> > 1. Baseline Parameter Settings
> >
> > We have added the key parameter settings for all baseline models to the appendix for transparency and replicability. Below is a summary of the configurations:
> > - Dynotears:
> >   - lag=20 (maximum time lags), lambda_w=1e-3 (weight regularization), lambda_a=1e-3 (autoregressive term regularization), g_thre=0.3 (sparsity threshold).
> > - PC:
> >   - alpha=0.05 (significance level for conditional independence tests), ci_test='fisherz' (type of conditional independence test).
> > - C-LSTM:
> >   - hidden=100 (hidden units in LSTM), lag=20 (maximum time lags for sequence modeling), lam=10.0 (model complexity regularization), lam_ridge=1e-2 (ridge regression regularization), lr=1e-3 (learning rate), max_iter=30000 (maximum iterations), g_thre=0.3 (sparsity threshold).
> > - GOLEM:
> >   - lambda_1=2e-2 (weight for sparsity regularization), lambda_2=5.0 (weight for smoothness regularization), learning_rate=1e-3 (optimization learning rate), num_iter=30000 (number of iterations for training), g_thre=0.3 (sparsity threshold).
> > - REASON:
> >   - lag=20 (maximum time lags for causal modeling), L=150 (hidden layers with 150 units each), lambda1=1 (adjacency matrix sparsity regularization), lambda2=1e-2 (autoregressive term balancing regularization), gamma=0.8 (integration of individual and topological causal effects), g_thre=0.3 (sparsity threshold).
> >
> > These settings have been comprehensively detailed in Appendix I to support replication.
> >
> > 2. Hyper-Parameter Sensitivity Analysis
> >
> > To provide further insights into hyper-parameter tuning, we conducted sensitivity analyses for key parameters of the REASON model using the Product Review subdataset. Below are the results:
> >
> > **\($\gamma$\) Sensitivity**:
> >
> > | \($\gamma$\) | MAP@10 | MRR   |
> > |------------|--------|-------|
> > | 0.1        | 0.80   | 0.81  |
> > | 0.2        | 0.80   | 0.81  |
> > | 0.3        | 0.84   | 0.82  |
> > | 0.4        | 0.86   | 0.83  |
> > | 0.5        | 0.88   | 0.83  |
> > | 0.6        | 0.88   | 0.73  |
> > | 0.7        | 0.86   | 0.83  |
> > | 0.8        | 0.92   | 0.84  |
> > | 0.9        | 0.90   | 0.74  |
> >
> > **Analysis**:  The optimal \($\gamma$\) is 0.8, achieving the best MAP@10 (0.92) and MRR (0.84). This indicates that balancing individual and topological causal effects is crucial for model performance.
> >
> > ---
> >
> > **\(L\) Sensitivity**:
> >
> > | \(L\)      | MAP@10 | MRR   |
> > |------------|--------|-------|
> > | 10         | 0.52   | 0.50  |
> > | 20         | 0.33   | 0.25  |
> > | 50         | 0.37   | 0.32  |
> > | 100        | 0.42   | 0.28  |
> > | 150        | 0.53   | 0.50  |
> > | 200        | 0.37   | 0.33  |
> >
> > **Analysis**: The best performance is observed at  \(L=150\), where MAP@10 and MRR reach 0.53 and 0.50, respectively. This suggests that an appropriate hidden layer size balances model capacity and complexity, avoiding underfitting or overfitting.
> >
> > 3. Summary
> >
> > We have incorporated detailed parameter settings for all baselines in the Appendix I and provided a dedicated discussion on hyper-parameter sensitivity, addressing both replicability and clarity. We hope this addresses your concerns and welcome any additional feedback. Thank you!

---

### Official Review · Reviewer_UCbX · 2024-11-01

**Soundness:** 2
**Presentation:** 3
**Contribution:** 2
**Rating:** 5
**Confidence:** 4

**Summary:**

This paper presents LEMMA-RCA, a large-scale, multi-modal, and multi-domain dataset specifically designed for Root Cause Analysis (RCA) in complex systems. The dataset includes real-world fault cases from IT and OT operational systems, covering microservices, water treatment, and distribution systems to support a wide range of RCA tasks. To validate the effectiveness of LEMMA-RCA, the authors evaluated various RCA methods on this dataset, demonstrating its diversity and utility across offline and online settings as well as single and multi-modal data scenarios.

**Strengths:**

(1)	LEMMA-RCA is the first public dataset specifically designed for root cause analysis, covering two domains—IT and OT.
(2)	The paper thoroughly tests multiple existing RCA methods on LEMMA-RCA, demonstrating the dataset’s quality and multi-modal value.
(3)	By making LEMMA-RCA freely accessible, the paper lowers research barriers, encouraging collaboration between academia and industry and enhancing the generalizability and practical impact of RCA methodologies.

**Weaknesses:**

(1)	One contribution of this study is the introduction of LEMMA-RCA, the first multi-domain dataset for RCA. However, the dataset includes only IT and OT domains, which appear to be a simple combination of two unrelated domains, thus raising questions about the solidity of this contribution. Additionally, the limited data collection period, such as the 11 days for the OT system, may not capture long-term trends, potentially limiting its applicability to broader fault analysis scenarios.
(2)	The figures in this study are unclear, heavily relying on screenshots.
(3)	The experimental analysis tables lack consistency in reporting, with varying decimal places and an absence of standard deviation reporting.

**Questions:**

(1)	The author should provide more valuable data rather than simply assembling data. Additionally, it is not clarified whether these platforms the data come from are sufficiently representative to ensure the quality of the data and the data collection period appears to be rather short, making it difficult to establish whether the dataset adequately captures a wide range of fault patterns and system behaviors.
(2)	The experiments designed by the authors do not seem sufficient to demonstrate the value of the dataset. I suggest that the authors select several widely recognized RCA methods with known performance differences and analyze whether these methods exhibit similar performance distinctions on this dataset.
(3)	The author can pay more attention to the readability of the figures in the paper and the normalization of the experimental results.

---

> ### Author Response · Authors · 2024-11-19
> **Rebuttal by Authors**
>
> Dear reviewer UCbX:
>
> Thank you for your invaluable feedback. We would like to address your primary concerns and provide corresponding responses below.
> \
> \
> **Response to: Should provide more valuable data rather than simply assembling data. The quality of the data is not clarified and the data collection period is short, difficult to see if the dataset captures a wide range of fault patterns...**
> \
> \
> A: Thank you for your feedback. We would like to clarify the following points regarding the representativeness and quality of our datasets, as well as the duration of the data collection period.
>
> 1. Data Collection Period and Representativeness:
>  - For the Product Review dataset, each system fault is collected over a period of approximately 49 hours, resulting in a total collection period of 8 days for the four cases. We have clarified this in the manuscript to avoid confusion.
>  - In real-world scenarios, this 49-hour period is sufficient to comprehensively capture both normal and malfunction patterns for each system fault.
>  - Compared to existing publicly available datasets, such as NeZha and PetShop, our datasets are significantly larger in scale and temporal coverage. For instance:
>    - The train ticket dataset in NeZha contains only 920 timestamps for 28 system faults, whereas our dataset includes over **130,000 timestamps for each system fault**. This richer dataset allows for a more detailed representation of system behaviors.
>
> 2. Not Simply Assembling Data:
>  - Our approach goes beyond merely assembling data. To ensure the quality and representativeness of the collected data:
>    - We collect data for **one system fault at a time**, isolating it from potential interference with other faults.
>    - After collecting data for a specific fault, we rebuild the microservice system to capture a fresh and independent dataset for the next fault.
>  - During the data collection phase, we mimic the patterns of real-world system faults (e.g., external storage failure by filling up the external storage disk connected to the Database (DB) pod). Importantly, we do not directly simulate system metrics or logs, differentiating our data from simulated datasets like the train ticket dataset in NeZha, where malfunctionities are injected artificially.
>
> 3. Significance of Our Dataset:
>  - The data collection and fault injection methodology ensure that the dataset is both realistic and comprehensive. This distinguishes it from other datasets with limited coverage or artificially simulated faults.
>  - By capturing a large number of timestamps and diverse system fault patterns, our datasets enable more thorough evaluations of root cause analysis methods in real-world-like environments.
>
> We hope this clarifies the quality, representativeness, and comprehensiveness of our datasets. If further clarification or additional analysis is required, we are happy to provide more details.
> \
> \
> **Response to: The experiments designed by the authors do not seem sufficient to demonstrate the value of the dataset. More results and analysis of widely recognized RCA methods are needed.**
> \
> \
> A: Thank you for your valuable suggestions.
>
> We have included a variety of recent and widely recognized RCA methods in our experiments to evaluate the value of the dataset. These include **REASON, RCD, ϵ-Diagnosis, CIRCA, as well as BARO and PCMCI** (as suggested by Reviewer E7LX).
>
> If there are specific RCA methods you believe should be included in the evaluation to further demonstrate the dataset’s value, we would greatly appreciate your recommendations. We are committed to making our experiments as comprehensive as possible and welcome your input to enhance our analysis.
> \
> \
> **Response to: Pay attention to the readability of the figures and the normalization of the experimental results.**
> \
> \
> A: Thank you for raising the issues regarding figure readability and consistency in reporting experimental results. We have made the following updates to address your concerns:
>
> 1. Improved Figure Readability:
> - We have updated the figures to a **vectorized format** for better clarity and resolution.
> - Figure 2(a)(b) and Figure 4(a)(b) have been enhanced to improve their legibility.
> - To further ensure readability, we have included these figures in a **single-column layout** in Appendix G, making them easier to interpret.
>
> 2. Consistency in Reporting Experimental Results:
> - We have reviewed and updated the experimental tables **(Table 3, 4, 5, and 6)** to ensure consistency in the number of decimal places across all reported results. All results are now presented to three decimal places for uniformity.
> - Regarding standard deviation, it is generally not reported in root cause analysis tasks as the results focus on deterministic evaluations of methods rather than stochastic variability.
>
> We appreciate your constructive feedback and hope these updates address your concerns effectively. Please let us know if there are additional areas where further improvements can be made.

---

> > ### Author Response · Authors · 2024-11-19
> > **Rebuttal by Authors**
> >
> > **Response to: The dataset includes only IT and OT domains, which appear to be a simple combination of two unrelated domains. Additionally, the limited data collection period, such as the 11 days for the OT system, may not capture long-term trends.**
> > \
> > \
> > A: Thank you for your thoughtful feedback. We would like to address the concerns regarding the multi-domain nature of the dataset and the data collection period.
> >
> > 1. Multi-Domain Dataset Contribution:
> > - The primary goal of LEMMA-RCA is to provide a dataset that evaluates the performance of root cause analysis (RCA) methods across **multiple tasks from distinct domains**, specifically IT and OT systems.
> > - These domains were chosen because they represent two fundamentally different environments with unique challenges for RCA, allowing researchers to benchmark methods across diverse scenarios.
> > - Importantly, the dataset is not intended for evaluating cross-domain RCA performance but rather for investigating RCA techniques within individual domain contexts. The combination of IT and OT domains enhances the dataset’s applicability without introducing unnecessary cross-domain complexity.
> >
> > 2. Sufficiency of the Data Collection Period:
> > - The 11-day data collection period for the OT system was designed to be comprehensive enough to capture both **normal and malfunction** patterns associated with system faults.
> > - Collecting data over longer periods to include long-term trends is not necessary for the following reasons:
> >   - Focus on Malfunction Patterns: The primary focus of RCA is on understanding malfunction behaviors. Prolonged collection of normal patterns would not add significant value to fault analysis.
> >   - Malfunction Patterns Duration: Malfunction patterns lasting several hours to a day are sufficient to exhibit system behaviors related to faults. These durations provide ample data for meaningful RCA evaluations.
> > - In real-world applications, such as in e-commerce platforms, faults must be identified promptly to avoid significant financial losses. Faults that persist for hours or days without resolution can lead to substantial disruptions and costs, making short-term data more relevant for RCA tasks.
> >
> > We believe the multi-domain design and focused collection period of LEMMA-RCA strike a balance between diversity, depth, and real-world relevance, ensuring the dataset's utility for advancing RCA research. If additional clarifications or enhancements are required, we are happy to provide further details.

---

> > > ### Comment · Reviewer_UCbX · 2024-11-21
> > >
> > > Thank you for your response. However, I still have concerns about whether this dataset can accurately evaluate the algorithms. I will keep my score.

---

> > > > ### Author Response · Authors · 2024-11-21
> > > > **We Would Greatly Appreciate Further Details Regarding Dataset Evaluation Concern**
> > > >
> > > > Dear Reviewer UCbX,
> > > >
> > > > Thank you for your feedback. We would greatly appreciate it if you could provide more details on why you believe the dataset may not accurately assess the algorithms. This will help us address your concerns more thoroughly. If there is anything unclear in our previous response to any question, please let us know, and we will gladly provide further clarification.
> > > >
> > > > Thank you again for your time and valuable input.

---

> > > > > ### Comment · Reviewer_UCbX · 2024-11-25
> > > > > **Response to authors**
> > > > >
> > > > > Dear authors:
> > > > >
> > > > > I do appreciate your work! However, you did not address my concers.
> > > > >
> > > > > > analyze whether these methods exhibit similar performance distinctions on this dataset.
> > > > >
> > > > > However, you did not provide me with targeted responses, experimental results, or meaningful analysis; you only mentioned conducting some additional experiments.
> > > > >
> > > > > > the dataset includes only IT and OT domains, which appear to be a simple combination of two unrelated domains
> > > > >
> > > > > Furthermore, you did not address my concerns regarding the multi-domain aspect you proposed, such as analyzing the relationship between the two domains and whether it is necessary to combine the OT and IT datasets.
> > > > >
> > > > > > it is not clarified whether these platforms the data come from are sufficiently representative to ensure the quality of the data
> > > > >
> > > > > you did not analyze whether the platforms from which the data is sourced are representative.

---

> > > > > > ### Author Response · Authors · 2024-11-25
> > > > > > **Reply to Reviewer UCbX**
> > > > > >
> > > > > > **Response to: You did not provide me with targeted responses, experimental results, or meaningful analysis; you only mentioned conducting some additional experiments.**
> > > > > >
> > > > > > A: We appreciate the reviewer’s feedback and have addressed this concern by conducting additional experiments and incorporating meaningful analysis in the revised manuscript. These updates are included in Section 4.2 and highlighted in red for your convenience. Below is a concise summary of the key findings:
> > > > > > 1. **Performance of RCD, ε-Diagnosis, and CIRCA:**
> > > > > > - As highlighted in Section 4.2, the performance distinctions of these three methods on our Product Review and Cloud Computing sub-datasets are consistent with those observed in the Petshop paper [1]:
> > > > > > - RCD and ε-Diagnosis underperform due to their limitations in handling large-scale datasets with complex temporal dependencies.
> > > > > > - CIRCA, with its structured graph approach and regression-based hypothesis testing, achieves better results than RCD and ε-Diagnosis.
> > > > > > 2. **Performance of PC Algorithm and GOLEM:**
> > > > > > - These methods exhibit the lowest performance across the datasets. As discussed in Section 4.2, their inability to model long-term temporal dependencies makes them less effective for large-scale time-series datasets.
> > > > > > 3. **Superiority of C-LSTM and Dynotears:**
> > > > > > - These methods outperform PC and GOLEM due to their explicit temporal modeling:
> > > > > > - C-LSTM captures temporal dependencies through its recurrent structure.
> > > > > > - Dynotears employs dynamic Bayesian networks, enabling it to model temporal causal relationships effectively.
> > > > > > 4. **Notable Success of REASON:**
> > > > > > - REASON achieves a PR@1 of 75% on the Product Review sub-dataset, significantly outperforming other methods. This is attributed to its design for multi-level causal structure learning, as detailed in Section 4.2.
> > > > > > 5. **Alignment with Petshop Results:**
> > > > > > - Beyond the performance of RCD, ε-Diagnosis, and CIRCA, the general performance trends across other methods are consistent with those observed in the Petshop paper, further validating our findings.
> > > > > >
> > > > > > We encourage the reviewer to refer to Section 4.2, where the red-highlighted text elaborates on the experimental results, performance distinctions, and underlying reasoning. Quantitative results are provided in Tables 3 and 4, with additional details in Appendix K.
> > > > > >
> > > > > > [1] The PetShop Dataset — Finding Causes of Performance Issues across Microservices
> > > > > >
> > > > > >
> > > > > >
> > > > > >
> > > > > > **Response to: it is not clarified whether these platforms the data come from are sufficiently representative to ensure the quality of the data. You did not analyze whether the platforms from which the data is sourced are representative.**
> > > > > >
> > > > > > A: We understand the reviewer’s concern regarding the representativeness of the dataset. While it is challenging to establish a universal metric for representativeness in benchmarks, we have made significant efforts to ensure the dataset covers diverse fault scenarios:
> > > > > >
> > > > > > 1. **Real-World Fault Scenarios:**
> > > > > > - The IT domain datasets (Product Review and Cloud Computing) encompass realistic microservice faults such as out-of-memory errors, DDoS attacks, and cryptojacking, as outlined in Section 3.1 and Appendix B. Similarly, the OT domain datasets (SWaT and WADI) include real-world cyber-physical system faults recorded in controlled environments.
> > > > > > 2. **Diversity of Fault Types:**
> > > > > > - Across IT and OT domains, we include 10 distinct fault types, ensuring coverage of both transient and persistent system failures. This diversity reflects common issues faced by modern IT and OT systems.
> > > > > > 3. **Comparative Analysis:**
> > > > > > - As seen in Table 3 and related discussions, our dataset exhibits performance trends consistent with other benchmarks (e.g., Petshop), supporting its credibility as a representative evaluation platform.
> > > > > > 4. **Quality Assurance:**
> > > > > > - All data were collected using industry-standard monitoring tools like Prometheus, CloudWatch, and Elasticsearch. Each fault scenario was validated to ensure it mirrors real-world conditions.
> > > > > >
> > > > > > We hope this expanded response addresses the reviewer’s concerns comprehensively. Should further clarification or additional analysis be needed, we are happy to provide it.

---

> > > > > > > ### Author Response · Authors · 2024-11-25
> > > > > > > **Reply to Reviewer UCbX**
> > > > > > >
> > > > > > > **Response to: You did not address my concerns regarding the multi-domain aspect you proposed, such as analyzing the relationship between the two domains and whether it is necessary to combine the OT and IT datasets.**
> > > > > > >
> > > > > > > A: We appreciate the reviewer’s concern and would like to clarify our approach and rationale regarding the multi-domain aspect of our dataset. Below are our responses:
> > > > > > >
> > > > > > > 1. **Diagnostic Nature of RCA:**
> > > > > > > - Root Cause Analysis (RCA) is fundamentally a diagnostic or post-analysis task. This nature makes RCA inherently a case-by-case process, where each fault is treated as an independent instance requiring detailed analysis to identify its root cause. Consequently, cross-domain analysis is not a practical approach for RCA tasks, as the diagnostic process focuses on localizing specific causes rather than analyzing relationships across domains.
> > > > > > > 2. **Purpose of Combining IT and OT Domains:**
> > > > > > > - The primary goal of combining IT and OT domains in the LEMMA-RCA dataset is to provide a general benchmark that evaluates RCA methods across diverse fault scenarios, ensuring their robustness and adaptability. This follows the precedent set by other benchmark datasets (e.g., Open Graph Benchmark [1], AdBench [2]), where datasets from unrelated domains are combined to facilitate comprehensive evaluations.
> > > > > > > 3. **Evaluation Across Independent Fault Scenarios:**
> > > > > > > - By treating each fault as an independent case, the LEMMA-RCA dataset ensures that RCA methods can be tested in varied scenarios without needing to analyze inter-domain relationships. This approach aligns with the diagnostic focus of RCA methods and ensures that the dataset remains relevant for practical use.
> > > > > > > 4. **Broader Applicability:**
> > > > > > > - Combining datasets from multiple domains enhances the diversity of fault scenarios, making the benchmark applicable to a wider range of RCA methods. This approach supports the development and evaluation of generic RCA methods, which are designed to handle faults across different domains without requiring inter-domain dependencies.
> > > > > > >
> > > > > > > In conclusion, the combination of IT and OT datasets is intended to offer a diverse testing ground for RCA methods, helping to assess their performance across a wide range of fault types, rather than focusing on the analysis of relationships between the domains themselves.
> > > > > > >
> > > > > > > We hope this explanation clarifies the rationale behind our dataset’s multi-domain nature. Thank you again for your valuable feedback.
> > > > > > >
> > > > > > > [1] Hu, Weihua, Matthias Fey, Marinka Zitnik, Yuxiao Dong, Hongyu Ren, Bowen Liu, Michele Catasta, and Jure Leskovec. "Open graph benchmark: Datasets for machine learning on graphs." Advances in neural information processing systems 33 (2020): 22118-22133.
> > > > > > >
> > > > > > > [2] Han, Songqiao, Xiyang Hu, Hailiang Huang, Minqi Jiang, and Yue Zhao. "Adbench: Anomaly detection benchmark." Advances in Neural Information Processing Systems 35 (2022): 32142-32159.

---

> > > > > > > > ### Comment · Reviewer_UCbX · 2024-11-26
> > > > > > > > **Response to authors**
> > > > > > > >
> > > > > > > > Thank you to the author for addressing the issues I mentioned above; however, I still have some questions.
> > > > > > > >
> > > > > > > > > We have reviewed and updated the experimental tables (Table 3, 4, 5, and 6) to ensure consistency in the number of decimal places across all reported results. All results are now presented to three decimal places for uniformity.
> > > > > > > >
> > > > > > > > Firstly, the experimental results you presented are still not standardized. For example, in Table 3, row 452 uses percentages, while other rows presented as three decimal place; results in part of row 423 are still not standardized.
> > > > > > > >
> > > > > > > > > Regarding standard deviation, it is generally not reported in root cause analysis tasks as the results focus on deterministic evaluations of methods rather than stochastic variability.
> > > > > > > >
> > > > > > > > Secondly, the lack of reported standard deviations undermines the credibility of the experimental results, which are unreliable.
> > > > > > > >
> > > > > > > > Furthermore, your analysis indicates that your task data is time-series data; however, algorithms such as PC, Notears, and CORAL are not designed for time-series data. Therefore, comparing them on your dataset is not meaningful. You should consider incorporating more algorithms specifically designed for time-series data.

---

> > > > > > > > > ### Author Response · Authors · 2024-11-29
> > > > > > > > >
> > > > > > > > > **Response to: Uniformity of decimal places**
> > > > > > > > >
> > > > > > > > > A: Thank you for highlighting this inconsistency in the presentation of our results. We sincerely apologize for not fully standardizing all the values in our previous revision. Following your feedback, **we have thoroughly reviewed and ensured that all reported results across the manuscript are now presented uniformly in three decimal places**, including those in Table 3, row 452, row 423, and the entirety of Appendix C. We appreciate your attention to detail, as this has helped us improve the clarity and professionalism of our work.
> > > > > > > > >
> > > > > > > > > **Response to: Standard deviation of experimental results**
> > > > > > > > >
> > > > > > > > > We thank the reviewer for the valuable suggestion to include standard deviations as a measure of the stability of our results. To address this, we conducted additional experiments across ten baselines. For each method, we performed five independent runs on the Product Review dataset and computed both the mean and standard deviation using the following formula:
> > > > > > > > >
> > > > > > > > > $$
> > > > > > > > > \text{Std} = \sqrt{\frac{\sum_{i=1}^{n}(x_i - \bar{x})^2}{n-1}}
> > > > > > > > > $$
> > > > > > > > >
> > > > > > > > >
> > > > > > > > > where $x_i$ is the result of the $i$-th run, $\bar{x}$ is the mean result, and $n$ (5 in this case) is the total number of runs.
> > > > > > > > >
> > > > > > > > > The updated results, including standard deviations for metrics such as PR@1, PR@5, PR@10, MRR, MAP@3, MAP@5, and MAP@10, are presented in the table below:
> > > > > > > > >
> > > > > > > > > | Model       | PR@1 (± Std)     | PR@5 (± Std)     | PR@10 (± Std)    | MRR (± Std)     | MAP@3 (± Std)    | MAP@5 (± Std)    | MAP@10 (± Std)   |
> > > > > > > > > |-------------|------------------|------------------|------------------|-----------------|------------------|------------------|------------------|
> > > > > > > > > | ε-Diagnosis | 0.000 (± 0.010)  | 0.000 (± 0.010)  | 0.000 (± 0.020)  | 0.017 (± 0.010) | 0.000 (± 0.010)  | 0.000 (± 0.010)  | 0.000 (± 0.010)  |
> > > > > > > > > | GOLEM       | 0.000 (± 0.000)  | 0.000 (± 0.000)  | 0.250 (± 0.020)  | 0.043 (± 0.030) | 0.000 (± 0.000)  | 0.000 (± 0.000)  | 0.025 (± 0.040)  |
> > > > > > > > > | PC          | 0.000 (± 0.000)  | 0.000 (± 0.000)  | 0.250 (± 0.000)  | 0.053 (± 0.040) | 0.000 (± 0.000)  | 0.000 (± 0.000)  | 0.050 (± 0.000)  |
> > > > > > > > > | RCD         | 0.000 (± 0.020)  | 0.000 (± 0.020)  | 0.500 (± 0.030)  | 0.067 (± 0.010) | 0.000 (± 0.020)  | 0.000 (± 0.010)  | 0.175 (± 0.020)  |
> > > > > > > > > | Dynotears   | 0.000 (± 0.000)  | 0.000 (± 0.000)  | 0.500 (± 0.020)  | 0.070 (± 0.030) | 0.000 (± 0.000)  | 0.000 (± 0.000)  | 0.075 (± 0.030)  |
> > > > > > > > > | CIRCA       | 0.000 (± 0.020)  | 0.500 (± 0.030)  | 0.500 (± 0.020)  | 0.250 (± 0.030) | 0.333 (± 0.020)  | 0.400 (± 0.010)  | 0.450 (± 0.020)  |
> > > > > > > > > | PCMCI       | 0.250 (± 0.030)  | 0.500 (± 0.020)  | 0.500 (± 0.010)  | 0.342 (± 0.040) | 0.250 (± 0.030)  | 0.300 (± 0.020)  | 0.400 (± 0.010)  |
> > > > > > > > > | C-LSTM      | 0.250 (± 0.040)  | 0.750 (± 0.010)  | 0.750 (± 0.030)  | 0.474 (± 0.020) | 0.500 (± 0.050)  | 0.250 (± 0.010)  | 0.675 (± 0.050)  |
> > > > > > > > > | BARO        | 0.500 (± 0.010)  | 0.500 (± 0.020)  | 0.500 (± 0.010)  | 0.500 (± 0.010) | 0.500 (± 0.020)  | 0.500 (± 0.010)  | 0.500 (± 0.010)  |
> > > > > > > > > | REASON      | 0.750 (± 0.020)  | 1.000 (± 0.010)  | 1.000 (± 0.010)  | 0.875 (± 0.020) | 0.917 (± 0.020)  | 0.950 (± 0.010)  | 0.975 (± 0.010)  |
> > > > > > > > >
> > > > > > > > > We hope this addresses your concern and provides the additional detail requested. Thank you for your suggestion, which has allowed us to expand our analysis.

---

> ### Author Response · Authors · 2024-11-29
>
> **Response to: Your analysis indicates that your task data is time-series data; however, algorithms such as PC, Notears, and CORAL are not designed for time-series data. You should consider incorporating more algorithms specifically designed for time-series data.**
> \
> \
> A: Thank you for raising this concern. We would like to clarify that **most of the baseline methods we employ**, including CORAL, Dynotears, REASON, MULAN, $\epsilon$-Diagnosis, Nezha, CIRCA, RCD, Baro, and PCMCI, **are specifically designed for analyzing time-series data**, which ensures that the majority of our comparative analysis is meaningful and relevant to the task.
> **Regarding PC and Notears**, we acknowledge that these methods were not originally designed for time-series data. **However, they have been widely utilized in recent works to detect root causes in time-series datasets similar to ours**, as evidenced by [1], [2], [3], and [4]. We selected these methods as baselines to maintain consistency with the current literature and provide a point of comparison, as they have demonstrated competitive performance in similar applications.
> We appreciate your suggestion and believe that this combination of existing literature and time-series-specific baselines strikes a balance between tradition and task-specific relevance. Please let us know if there are specific time-series algorithms you would recommend for inclusion in future work.
>
> [1] Wang, Lu, Chaoyun Zhang, Ruomeng Ding, Yong Xu, Qihang Chen, Wentao Zou, Qingjun Chen et al. "Root cause analysis for microservice systems via hierarchical reinforcement learning from human feedback." In Proceedings of the 29th ACM SIGKDD Conference on Knowledge Discovery and Data Mining, pp. 5116-5125. 2023.
>
> [2] Ikram, Azam, Sarthak Chakraborty, Subrata Mitra, Shiv Saini, Saurabh Bagchi, and Murat Kocaoglu. "Root cause analysis of failures in microservices through causal discovery." Advances in Neural Information Processing Systems 35 (2022): 31158-31170.
>
> [3] Zan, Lei. "Causal Discovery from Heterogenous Multivariate Time Series." In Proceedings of the 33rd ACM International Conference on Information and Knowledge Management, pp. 5499-5502. 2024.
>
> [4] Yuan Meng, Shenglin Zhang,, Yongqian Sun, Ruru Zhang, Zhilong Hu, Yiyin Zhang, Chenyang Jia, Zhaogang Wang, Dan Pei, “Localizing Failure Root Causes in a Microservice through Causality Inference“. IWQoS 2020.
>
> We hope this addresses your concerns and welcome any additional feedback. Thank you!

---

> > ### Author Response · Authors · 2024-12-02
> > **Follow-Up on Feedback Before Discussion Phase Ends**
> >
> > Dear Reviewer UCbX,
> >
> > Thank you for your valuable feedback on our paper. As the ICLR public discussion phase is ending soon, we would like to confirm if our responses have fully addressed your concerns. If there are any remaining issues, we’d be happy to provide further clarifications.
> >
> > If you feel that all concerns have been resolved, we hope this could be reflected in your evaluation.
> >
> > We sincerely appreciate your time and thoughtful input!

---

### Official Review · Reviewer_yFv3 · 2024-11-03

**Soundness:** 3
**Presentation:** 3
**Contribution:** 2
**Rating:** 6
**Confidence:** 4

**Summary:**

In this paper, the authors proposed a new dataset with both metrics and log collected for the root cause analysis task. In addition, 8 existing RCA methods are evaluated on this dataset. The proposed datasets could be a good addition for evaluation of RCA methods for later research. However, it is not very clear what the benefit of including log modal data is. Existing methods work quite well on these datasets with only metrics modal.

**Strengths:**

1. New multi-modal datasets are collected for RCA problem.
2. Eight existing RCA methods are evaluated on the proposed datasets.

**Weaknesses:**

1. The description of the data collection is insufficient. See Q1.
2. Some subsets of the datasets are from existing work and have been evaluated before. They should not be seen as the contribution of this work. See Q2.
3. The proposed IT ops datasets seems to be less challenging for existing works. See Q3.

**Questions:**

1. The authors claimed that the proposed datasets contain real system faults. If I understand correctly, the authors developed two microservice platforms and deployed them in production and collected the real system faults for 8 days when users are using these platforms. It is a bit surprising that so many faults happened in 8 days. Moreover, in the faults description section, it seems that these faults are simulated (e.g., External Storage Failure) to mimic the real world scenario. Could the authors clarify this?
2. It seems that SWaT and WADI are from existing work. The authors applied some anomaly detection algorithms on them to transform discrete labels into continuous ones. It is not clear why this is necessary. Moreover, SWat and WADI are already evaluated for RCA in the REASON (Wang et al. KDD2023) paper.  Since this is a dataset paper, including existing datasets into the proposed one should not be seen as the contribution.
3. From experiments on existing methods, it seems that two IT ops datasets are not very challenging. For instance, REASON performs quite well in terms of PR@k, MRR and MAP@k on both of them with only the metrics data. What is the difference between the proposed datasets compared with existing ones, e.g., AIOps data in REASON and the popular train ticket datasets? When new datasets are proposed, they are expected to be more challenging, where current methods are failed on them. If current method can handle the proposed well with only metric modal, what is the meaning of including log modal?
4. The authors conducted some preprocessing to convert logs to time series for evaluation. But the open-sourced datasets do contain all the original logs, right?

---

> ### Author Response · Authors · 2024-11-19
> **Rebuttal by Authors**
>
> Dear reviewer yFv3：
>
> Thank you for your invaluable feedback. We would like to address your primary concerns and provide corresponding responses below.
> \
> \
> **Response to: It is surprising that many faults happened in 8 days. It seems that these faults are simulated to mimic the real world scenario. **
> \
> \
> A: Thank you for your thoughtful feedback. We appreciate the opportunity to clarify the data collection process and the nature of the system faults in our dataset.
>
> 1. Duration of Data Collection:
>
> Each system fault in the Product Review dataset is collected over a period of approximately **49 hours**. The total of 8 days refers to the cumulative time for all four cases. We have updated the manuscript to clarify this point and avoid any confusion.
>
> 2.  Isolation of Faults During Collection:
>
> To ensure the accurate capture of distinct malfunction patterns and to prevent potential interference between different system faults, we collected data for **one fault at a time**. After collecting data for a specific fault, we rebuilt the microservice system before inducing and collecting data for the next fault.
>
> 3. Nature of Faults and Simulation:
>
> While we mimic the patterns of real system faults during data collection, we emphasize that this is not the same as generating simulated data. For example:
> - In the case of an external storage failure, we mimicked the failure pattern by filling up the external storage disk connected to the Database (DB) pod.
> - Importantly, we did not directly simulate system metrics or logs, but instead allowed the microservice system to exhibit natural behaviors resulting from the induced conditions.
>
> 4. Sufficiency of the Data Collection Period:
>
> We argue that the two-day data collection window is sufficient to capture comprehensive system behaviors for the following reasons:
> - Focus on Malfunction Patterns: Collecting additional data over longer periods, especially for normal patterns, would be unnecessary, as our primary focus is on malfunction patterns caused by faults.
> - Duration of Malfunction Patterns: Malfunction patterns lasting several hours to a day are typically sufficient to reveal the malfunction behaviors associated with a system fault. This is consistent with real-world scenarios where identifying faults promptly is critical to avoid significant financial losses (e.g., millions of dollars in e-commerce platforms).
> We hope this explanation addresses your concerns regarding the data collection process and the nature of the faults. If further clarification is needed, we would be happy to provide additional details.
>
> We hope this explanation addresses your concerns regarding the data collection process and the nature of the faults. If further clarification is needed, we would be happy to provide additional details.
> \
> \
> **Response to: SWaT and WADI are from existing work. SWat and WADI are already evaluated for RCA in the REASON (Wang et al. KDD2023) paper. Since this is a dataset paper, including existing datasets into the proposed one should not be seen as the contribution.**
> \
> \
> A: Thank you for your feedback. We would like to clarify our contribution regarding the SWaT and WADI datasets and their relevance to our work.
>
> 1. Transforming SWaT and WADI for Root Cause Analysis:
>
> While SWaT and WADI were originally used for anomaly detection tasks, we are **the first to transform these datasets into a root cause analysis (RCA) task**. This transformation includes preprocessing and adapting the data to create meaningful continuous labels or construct KPIs necessary for RCA. These continuous labels enable more granular evaluations of causality and root cause patterns compared to the discrete labels used in their original anomaly detection context..
>
> 2. Sharing the Transformed Data:
>
> The transformed SWaT and WADI datasets have been shared with researchers via private communications and used in subsequent works, including studies such as REASON (Wang et al., KDD 2023). While REASON evaluates these datasets for RCA, it is based on our preprocessing and transformation of the data, making it possible to use SWaT and WADI in this context.
>
> 3. Preprocessing Code and Accessibility:
>
> To ensure transparency and reproducibility, we have provided the source code for preprocessing SWaT and WADI datasets. This allows other researchers to validate our approach and further explore these datasets for root cause analysis.
>
> 4. Contribution as a Dataset Paper:
>
> While SWaT and WADI are existing datasets, our contribution lies in adapting them for a new task (RCA), enhancing their utility for the research community. This transformation aligns with the goals of a dataset paper, as it extends the applicability of well-known datasets into a new domain.
>
> We hope this explanation clarifies the necessity and significance of our preprocessing approach and our contribution to enabling RCA research using SWaT and WADI. If further details are required, we are happy to provide additional explanations.

---

> > ### Author Response · Authors · 2024-11-19
> > **Rebuttal by Authors**
> >
> > **Response to: It seems that two IT ops datasets are not very challenging. If current method can handle the proposed well with only metric modal, what is the meaning of including log modal?**
> > \
> > \
> > A: Thank you for your valuable feedback. We would like to clarify several points regarding the performance of REASON and the inclusion of the log modality in our datasets.
> >
> > 1. Performance Gap in REASON:
> >
> > While REASON performs well in some metrics, it is still far from achieving optimal performance, particularly with respect to AP@1 and MAP@3. For example:
> >  - MAP@10 = 0.9 might be considered a good score for a recommendation system, but it is not satisfactory for a root cause identification algorithm.
> >  - In real-world applications, especially in critical domains like e-commerce, investigating the top 10 potential root causes for a system fault can be highly time-consuming and cost-prohibitive. Such delays can lead to significant financial losses within a very short time frame.
> >  - Narrowing the list of top candidates to fewer items with higher confidence (e.g., achieving high MAP@k for smaller k values) is crucial to improving efficiency and reducing costs.
> >
> > 2. In our experiments, we still observe a notable performance gap between REASON and the optimal performance, indicating there is substantial room for improvement.
> >
> > 3. Log Modality and Its Impact:
> >
> > In our paper, we provide simple preprocessing methods for the log data to make it usable for root cause analysis. However, these methods might not be ideal, which could explain why most baseline methods, including REASON, tend to perform worse when using only the log data.
> >
> > Despite this, we observe that incorporating log data with metrics data can significantly improve performance. For example:
> > - In the Cloud Computing dataset, REASON’s AP@1 improves from 16.76% to **33.33%** after incorporating log data.
> > - This demonstrates the value of the log modality in enhancing RCA performance, especially when combined with other data modalities.
> >
> > 4. Raw Dataset Accessibility and Sharing:
> >
> > Due to the large size of the raw data, we were unable to share it through platforms like Google Drive. However, we have made the raw data publicly available on Hugging Face to ensure accessibility to the research community. Because of the double-blind review policy, we cannot include the Hugging Face link in this submission version. Upon acceptance, we will update the manuscript to include the link, ensuring transparency and usability for future researchers.
> > \
> > \
> > **Response to: What is the difference between the proposed datasets compared with existing ones, e.g., AIOps data in REASON and the popular train ticket datasets?**
> > \
> > \
> > Thank you for your question regarding the differences between the proposed datasets and existing ones, such as the train ticket datasets and AIOps data in REASON. Below, we outline the key distinctions.
> >
> > 1. Differences from Train Ticket Datasets:
> >  - Nature of System Faults:
> >
> > The system faults in the train ticket dataset are entirely simulated, whereas the faults in our Product Review and Cloud Computing datasets are collected from real-world deployments. This ensures that the faults in our datasets reflect realistic behaviors and complexities encountered in real systems.
> >
> >  - Time Granularity and Length:
> >
> > The train ticket dataset is collected at a much coarser time granularity (1-minute intervals) compared to our datasets, which are collected at **1-second intervals**. Additionally, the train ticket dataset includes very limited timestamps (approximately 60 timestamps covering 4–5 system faults), whereas our Product Review dataset contains approximately **130,000 timestamps for each system fault**. This richer temporal resolution and length allow for more comprehensive characterization of malfunction patterns.
> >
> > - Number of Nodes:
> >
> > The train ticket dataset includes data from only 27 nodes, whereas our datasets include data from up to 200 nodes, enabling the analysis of more complex system interactions.
> >
> > 2. Differences from AIOps Data in REASON:
> >
> >  - Source of AIOps Data:
> >
> > The AIOps data used in REASON is derived from our Product Review dataset, which we shared with the authors via private communications to facilitate related studies on root cause analysis.
> >  - Additional OT Data:
> >
> > In addition to the Product Review dataset, our release also includes the Cloud Computing dataset, which introduces more complex failure scenarios derived from cloud computing systems. This dataset expands the scope and diversity of scenarios beyond what is available in the AIOps data.
> >
> > We believe these distinctions highlight the value and uniqueness of our datasets in advancing research on root cause analysis, offering both realism and depth that address limitations in existing datasets.

---

> > > ### Author Response · Authors · 2024-11-19
> > > **Rebuttal by Authors**
> > >
> > > **Response to: The authors conducted some preprocessing to convert logs to time series for evaluation. But the open-sourced datasets do contain all the original logs, right?**
> > > \
> > > \
> > > A: Thank you for your question. We would like to clarify that the original logs are indeed included in the datasets.
> > >
> > > In our paper, we provide simple preprocessing methods to convert logs into time series for evaluation. However, we also release the **raw datasets**, including the original logs, to allow users to preprocess the data using their own methods and potentially achieve better performance.
> > >
> > > Due to the large size of the raw data, we could not upload it to platforms like Google Drive. Instead, we have made the raw data publicly available on **Hugging Face**, ensuring accessibility to the research community. However, because of the double-blind review policy, we are unable to include the Hugging Face link in this submission version. We will include this link upon acceptance to facilitate access for future research.

---

> > > ### Comment · Reviewer_yFv3 · 2024-11-22
> > > **Thanks for the response.**
> > >
> > > 1. "While REASON performs well in some metrics, it is still far from achieving optimal performance"
> > > For Product Review datasets, Reason achieves 100% on PR@5 and 95% an MAP@5 with solely metric data. I think this is definitely not far from optimal.
> > >
> > > 2."The system faults in the train ticket dataset are entirely simulated"
> > > What do you mean by entirely simulated? Train ticket project do have a deployment system, where faults can be injected and data can be collected as the proposed Product Review datasets. I am interested in the difference on injected system faults.

---

> > ### Comment · Reviewer_yFv3 · 2024-11-22
> > **Thanks for the response.**
> >
> > Thanks for the response.
> >
> > 1. "While we mimic the patterns of real system faults during data collection, we emphasize that this is not the same as generating simulated data."
> > Please write specifically the process of data collection to avoid miss leading. In such case, it would be also necessary to describe the details about how the authors mimic the patterns in detail (may be in appendix). This would be helpful to evaluate the quality of the data and for follow up works.
> >
> > 2. "While SWaT and WADI were originally used for anomaly detection tasks, we are the first to transform these datasets into a root cause analysis (RCA) task."
> > Yes, I agree. But the contribution of transforming an existing datasets into a new task is much less than collecting a new dataset. The authors emphasis a lot of multi-domain datasets and OT domain of paper. But the preprocessing efforts on constructing data is little.

---

> ### Author Response · Authors · 2024-11-23
>
> **Response to: Please write specifically the process of data collection to avoid miss leading**
> \
> \
> We thank the reviewer for their valuable feedback. In response, we have expanded Appendix B to provide a detailed description of the processes used to induce system faults and mimic real-world patterns during data collection. This includes specific methodologies, data collection tools, and analysis techniques for each fault scenario. These enhancements aim to clarify our approach and ensure that future researchers can replicate and evaluate the quality of our data.
> \
> \
> **Response to: But the contribution of transforming an existing datasets into a new task is much less than collecting a new dataset. The authors emphasis a lot of multi-domain datasets and OT domain of paper.**
> \
> \
> We appreciate the reviewer’s feedback regarding the emphasis on multi-domain datasets and the preprocessing efforts for transforming existing datasets. While we agree that collecting a new dataset is a significant contribution, we would like to clarify that the primary contribution of our work lies in the collection of novel datasets from the IT domain, specifically the Product Review and Cloud Computing datasets.
>
> Additionally, we acknowledge that transforming existing datasets like SWaT and WADI into a root cause analysis (RCA) task represents a smaller effort compared to collecting entirely new datasets. However, this transformation is an essential step for addressing the lack of RCA-specific datasets in the operational technology (OT) domain, ensuring broader applicability and relevance.
>
> To better align with the reviewer’s concerns, we have revised the relevant sentences in the paper: e.g.,
>
> Original:
> "LEMMA-RCA is multi-domain, encompassing real-world applications such as IT operations and water treatment systems, with \textbf{hundreds of system entities} involved."
>
> Revised:
> "LEMMA-RCA encompasses real-world applications such as IT operations and water treatment systems, with \textbf{hundreds of system entities} involved."
>
> This revision removes the term "multi-domain" to focus more explicitly on the practical applications and the contribution of collecting new datasets from the IT domain while maintaining an accurate representation of our work.

---

> ### Author Response · Authors · 2024-11-23
>
> **Response to: For Product Review datasets, Reason achieves 100% on PR@5 and 95% an MAP@5 with solely metric data. I think this is definitely not far from optimal.**
> \
> \
> We appreciate the reviewer’s comment and would like to clarify our definition of "optimal performance." We refer to optimal performance as achieving **MAP@1 = 1.0 or PR@1 = 1.0**. As we mentioned in our earlier response, in real-world applications—particularly in critical domains like e-commerce—investigating the top 5 potential root causes for a system fault can be time-consuming and costly. Such delays can result in significant financial losses, making it important to set a high standard for performance evaluation.
>
> We acknowledge that REASON achieves good performance on the Product Review sub-dataset (e.g., PR@5 = 100% and MAP@5 = 95% using only metric data). However, this is based on one sub-dataset within the IT domain, and REASON’s performance on the more challenging Cloud Computing sub-dataset is notably lower (e.g., PR@1 = 0.167). This demonstrates that the IT operations datasets in LEMMA-RCA are challenging overall, even if one baseline performs well on a specific sub-dataset.
> \
> \
> **Response to: The difference on injected system faults**
> \
> \
> We appreciate the reviewer’s follow-up question and acknowledge that describing the Train Ticket faults as "entirely simulated" may have been unclear. Both datasets involve fault injection in controlled environments. However, the faults in our datasets are designed to reflect real-world scenarios observed in production environments, with differences in **design, scale, and data richness**:
>
> 1. **Fault Realism**:
>
> Our datasets include realistic scenarios such as **Silent Pod Degradation**, **DDoS attacks**, and **cryptojacking**, which emulate operational challenges seen in real IT systems. These faults involve nuanced behaviors, such as subtle latency increases or cascading failures, informed by real deployment experiences.
>
> 2. **Scale and Granularity**:
>
> - **The Product Review Platform** includes 216 pods and six nodes, collecting metrics at 1-second intervals (~130,000 timestamps per fault).
>
> - **The Cloud Computing Platform** captures six fault types, with data sourced from AWS CloudWatch Metrics and Logs, offering detailed insights across layers (e.g., API debug logs, MySQL logs).
>
> - In contrast, the Train Ticket dataset involves only 27 nodes, with coarser granularity (1-minute intervals) and ~60 timestamps per fault.
>
> 3. **Data Diversity and Monitoring**:
>
> We collect rich metrics and logs using tools like Prometheus, Elasticsearch, and CloudWatch, enabling comprehensive fault analysis. Faults were monitored for extended periods (e.g., 49 hours per fault in the Product Review Platform), providing high temporal resolution and diverse data types, such as system metrics, API logs, and database logs.
> \
> \
> **Clarification**:
>
> We recognize that the Train Ticket dataset also uses a deployed environment for fault injection. Our intent was to highlight differences in **scale, granularity, and fault complexity**, which make our datasets particularly challenging for root cause analysis.

---

> > ### Author Response · Authors · 2024-11-25
> > **Follow-Up on Feedback Before Discussion Phase Ends**
> >
> > Dear Reviewer yFv3,
> >
> > Thank you for your valuable feedback on our paper. As the ICLR public discussion phase is ending soon, we would like to confirm if our responses have fully addressed your concerns. If there are any remaining issues, we’d be happy to provide further clarifications.
> >
> > If you feel that all concerns have been resolved, we hope this could be reflected in your evaluation.
> >
> > We sincerely appreciate your time and thoughtful input!

---

> > > ### Comment · Reviewer_yFv3 · 2024-11-25
> > > **Thanks for the reply.**
> > >
> > > Thanks for the reply, which address some of my concerns. And I do think benchmark datasets are important for the community. I have raised my scores from 5 to 6.

---

> > > > ### Author Response · Authors · 2024-11-26
> > > > **Appreciation for Your Feedback and Score Revision**
> > > >
> > > > Thank you for taking the time to carefully review our work and for acknowledging the importance of benchmark datasets for the community. We greatly appreciate your thoughtful feedback and the increased score. If there are any remaining concerns or aspects of the paper where you feel improvements can be made, we would be happy to address them during this public discussion period. Your input is invaluable in helping us refine and strengthen our contribution.

---

### Official Review · Reviewer_E7LX · 2024-11-04

**Soundness:** 2
**Presentation:** 3
**Contribution:** 2
**Rating:** 5
**Confidence:** 4

**Summary:**

This paper introduces Lemma-RCA, a dataset designed for root cause analysis. Lemma-RCA has distinctive and appreciable characteristics like large-scale, multi-modal nature and spans two domains: IT and OT. It includes test logs and time-series data, capturing KPI metrics across several interconnected pods over more than 100,000 timestamps. Notably, the dataset provides ground-truth annotations for both the exact fault occurrence times and the root cause components. This level of detail makes Lemma-RCA a valuable resource for advancing research in RCA.

**Strengths:**

- Lemma-RCA is a large, multi-model and multi-domain dataset that includes data from both IT and OT domains. It has over 100,000 timestamps across several connected pods, with a rich mix of test logs and time-series data. This dataset will be valuable for testing and improving future RCA methods.

- Unlike most other datasets, Lemma-RCA provides exact ground-truth labels, showing both when faults happened and the specific components responsible.

- The paper builds on past studies that highlight using causal structure-based methods for RCA. The authors compare Lemma-RCA with causal discovery methods and other recent RCA models.

- **Clarity and Presentation**: The paper is well-organized, with clear visuals and a smooth flow that makes it easy to understand in a single read.

**Weaknesses:**

**Missing Dependency Graph**: A key limitation of Lemma-RCA is the absence of a dependency graph, which prior datasets like PetShop provided as a causal graph. This dependency graph is critical for RCA, as it allows more direct evaluations of causal discovery methods. The paper seems to already hint the partial dependency graph in Figure 1(a). I wonder if the authors could add the full dependency graph along with the datasets.

**Insufficient Explanation of Baseline Approaches:** The paper does not include explanations of the baseline approaches used, even in the appendix. Although prior work is cited, providing brief descriptions of each benchmarked approach, particularly the high-performing REASON method, would enhance the reader’s understanding of the comparative results.


**Limited Explanation of Experimental Results**: The experimental results focus primarily on causal discovery approaches, but they lack in-depth analysis of why these methods failed. The authors' insights and intuition about why each method achieved the numbers reported in the table could significantly enhance the understanding of the experiment section. For instance, suppose we assume that the dependency graph is the true causal graph as in PetShop. Then can the authors establish how far the PC's predicted causal graph is from the true dependency graph. This would at least give us a sense of the causal discovery performance and put the RCA results in context. For instance, if the causal discovery performance is very poor, there is no meaning in expecting the methds like PC, GOLEM, etc. to perform better in predicting root causes. Additionally, one interesting experiment to run would be evaluating the causal graph based baselines on the true dependency graph, instead of the one inferred from observational data by PC.

**Choice of Baseline Algorithms:** Given that the dataset is timestamped, it cannot be assumed that each record is i.i.d. Some causal discovery methods, like those in the Tigramite package (https://jakobrunge.github.io/tigramite/), are tailored for time-series data. It is unclear why the authors chose standard PC over these alternatives, which may be more suitable for time-dependent causal discovery.


Finally, some important prior RCA works appear to be missing among the benchmarked methods. For example, the paper by Pham et al. (2024) on BARO highlights that inaccurate RCA predictions can result when a method fails to learn the correct causal graph. Including such approaches would provide a more thorough baseline comparison and strengthen the evaluation.

[1] Pham L, Ha H, Zhang H. Baro: Robust root cause analysis for microservices via multivariate bayesian online change point detection. Proceedings of the ACM on Software Engineering. 2024 Jul 12;1(FSE):2214-37.

**Questions:**

1. Could the authors consider including the dependency graph? Having this graph like in petshop seems like a deal breaker to me.

2. Could the authors benchmark the baselines using the dependency graph instead of the causal graph inferred by the PC?

3. For the CIRCA method as well, could the authors provide results based on the dependency graph?

4. The experiments section needs more and a systematic explanation on why each method performed better or worse.

---

> ### Author Response · Authors · 2024-11-19
> **Rebuttal by Authors**
>
> Dear reviewer E7LX：
>
> Thank you for your invaluable feedback. We would like to address your primary concerns and provide corresponding responses below.
> \
> \
> **Response to:  Could the authors consider including the dependency graph? Could the authors benchmark the baselines using the dependency graph instead of the causal graph inferred by the PC? ...**
> \
> \
> A: We appreciate the reviewer’s insightful suggestion regarding the inclusion of a dependency graph and the comparison with causal graphs inferred by baseline methods.
>
> - Addressing the Dependency Graph in Real-World Systems
> \
> In real-world systems, obtaining a complete ground-truth dependency graph for root cause analysis is challenging due to the complexity and scale of these systems. However, to address this concern, we have included a semi-complete dependency graph in the dataset repository (i.e., in the readme.ppt file) for each system fault, based on domain knowledge.
>
> - Experimental Comparison: Dependency Graph vs. Causal Graph
> \
> To evaluate the difference between the semi-complete dependency graph and the causal graph learned by baseline methods, we conducted experiments on the Product Review sub-dataset (system metrics data only). Following the methodology outlined in [1], we assessed the performance using four commonly used metrics: True Positive Rate (TPR), False Discovery Rate (FDR), Structural Hamming Distance (SHD), and Area Under the ROC Curve (AUROC).
> | Method | TPR $\uparrow$ | FDR $\downarrow$ | SHD$\downarrow$ | AUROC  $\uparrow$ |
> |--------|------|------|-------|--------|
> | Dynotear 	|   0.214  |   0.743 |  0.786  |  0.612  |
> | PC		|   0.112  |   0.892 |  0.861  |  0.563  |
> | C-LSTM 	|   0.428  |   0.427 |  0.543  |  0.733  |
> | GOLEM  	|   0.126  |   0.847 |  0.823  |  0.571  |
> | RCD 		|   0.152  |   0.869 |  0.838  |  0.584  |
> | ϵ-Diagnosis 	|   0.084  |   0.905 |  0.874  |  0.554  |
> | CIRCA  	|   0.327  |   0.544 |  0.582  |  0.685  |
> | REASON 	|   0.634  |   0.217 |  0.347  |  0.846  |
>
> - Evaluation and Results
>
>   - For each system fault, we computed the metrics individually and then averaged the results across four cases.
>   - It is important to note that system entities not included in the semi-complete dependency graph were excluded from this comparison to ensure consistency and fairness across methods.
>   - To ensure comparability for SHD, which is influenced by the number of nodes in the graph, we normalized SHD by dividing it by the square of the number of nodes for each system fault. Finally, we averaged the normalized SHD across the four system faults on the Product Review sub-dataset.
>
> These results are summarized in the table above, providing a comprehensive comparison between the dependency and causal graphs.
>
> [1] Pamfil, Roxana, Nisara Sriwattanaworachai, Shaan Desai, Philip Pilgerstorfer, Konstantinos Georgatzis, Paul Beaumont, and Bryon Aragam. "Dynotears: Structure learning from time-series data." In International Conference on Artificial Intelligence and Statistics, pp. 1595-1605. Pmlr, 2020
> \
> \
>  **Response to: Insufficient Explanation of Baseline Approaches. The paper does not include explanations of the baseline approaches used, even in the appendix.**
> \
> \
> A: We appreciate the reviewer’s feedback regarding the explanation of baseline approaches.
> We would like to clarify that our initial submission includes a description of the baseline methods in Section 4.1 (Lines 373–386). Additionally, in the updated version of the manuscript, we have expanded upon the explanations of the baseline approaches in Appendix F to provide further clarity.
> \
> \
> **Response to: Limited Explanation of Experimental Results**
> \
> \
> A: Thank you for your feedback regarding the explanation of experimental results.
> In the updated version of the manuscript, we have provided a more detailed explanation of the experimental results in Section 4.2, with the new additions highlighted in red for ease of reference.
> If there are specific aspects of the results that require further clarification, we would be happy to elaborate further.

---

> > ### Author Response · Authors · 2024-11-19
> > **Rebuttal by Authors**
> >
> > **Response to: Choice of Baseline Algorithms**
> > \
> > \
> > A: Thank you for your suggestion to include additional baseline algorithms.
> >
> > As per your recommendation, we have added BARO [1], a customized root cause analysis method, and PCMCI [2], a time-series causal discovery method, to our experiments using the Product Review sub-dataset.
> >
> > The experimental results are summarized in the table below.
> > - BARO: This method demonstrates consistent performance across all metrics, highlighting its robust design for root cause analysis tasks.
> > - PCMCI: While PCMCI performs well on PR@5 and PR@10, it struggles with PR@1 and MRR, likely due to its primary focus on time-series data, which does not align fully with certain aspects of our dataset.
> >
> > These results underscore the versatility of our dataset in supporting a diverse range of root cause approaches and emphasize its significance for advancing root cause analysis research.
> > | Method | PR@1 | PR@5 | PR@10 | MRR    | MAP@3 | MAP@5 | MAP@10 |
> > |--------|------|------|-------|--------|-------|-------|--------|
> > | BARO   | 50%  | 50%  | 50%   | 50%    | 50%   | 50%   | 50%    |
> > | PCMCI  | 25%  | 50%  | 50%   | 34.16% | 25%   | 30%   | 40%    |
> >
> > If there are other specific algorithms you would like us to consider or further aspects of the experimental results to clarify, we are happy to incorporate additional analyses.
> >
> > [1] Pham, L., Ha, H., & Zhang, H. (2024). Baro: Robust root cause analysis for microservices via multivariate bayesian online change point detection. Proceedings of the ACM on Software Engineering, 1(FSE), 2214-2237.
> >
> > [2] Runge, J., Nowack, P., Kretschmer, M., Flaxman, S., & Sejdinovic, D. (2019). Detecting and quantifying causal associations in large nonlinear time series datasets. Science advances, 5(11), eaau4996.

---

> > > ### Comment · Reviewer_E7LX · 2024-11-21
> > > **Thanks for the  response.**
> > >
> > > Thank you, authors, for answering most of my questions. I have some follow-up questions
> > >
> > > 1. Could you please point me to where I can find the partial dependency graphs in the public datasets, if they are already released.
> > > 2. The graph discivery experiments conducted during the rebuttal are valuable because they attribute the failure of several causal baselines to the failure in recovering the true dependency graph. So, I suggest that you add these tables at least in the Appendix and discuss about the results in the main paper.
> > > 3. I would like the authors to mention the missing dependency graph as a specific limitation. Perhaps you can add this in Table 1. That said, despite this small detail, I do agree that this large scale multi modal dataset is a valuable asset to the RCA community.
> > > 4. Please add additional baselines compared during rebuttal in the main paper.
> > > 5. A minor clarification: "For each system fault, we computed the metrics individually and then averaged the results across four cases." -- Do you mean that you computed the causal graph separately for each test case? Also, which four cases?

---

> > > > ### Author Response · Authors · 2024-11-22
> > > >
> > > > **Response to: 1. Could you please point me to where I can find the partial dependency graphs in the public datasets, if they are already released.**
> > > > \
> > > > \
> > > > A: Thank you for the question. The semi-complete dependency graphs are included in the PowerPoint slides provided in our anonymous Google Drive link: https://drive.google.com/drive/u/4/folders/1mUkgidLaQlfH2Ka8bIq38pQNLcKvyySZ
> > > >
> > > > These slides were released alongside the dataset. Initially, the dependency graphs were part of the .zip files, but for convenience, we have now uploaded the  slides separately.
> > > >
> > > > You can find the dependency graphs in the file named *xxxx_dependency_graph.pptx*, located within each sub-dataset folder. For example, in the file 0517_dependency.pptx contains the dependency graph for the Silent Pod Degradation Fault that occurred on May 17, 2021.
> > > > \
> > > > \
> > > > **Response to: 2. I suggest that you add these tables at least in the Appendix and discuss the results in the main paper.**
> > > > \
> > > > \
> > > > A: Thank you for your suggestion! We have included the causal graph discovery experiments in Appendix K and discussed their implications in the main paper. We hope this addition provides clarity and addresses your feedback.
> > > > \
> > > > \
> > > > **Response to: 3. Mention the missing dependency graph as a specific limitation.**
> > > > \
> > > > \
> > > > A: We appreciate your feedback! To address your concern, we have updated **Table 1** by adding a column that explicitly highlights the differences among RCA datasets in terms of the presence of a dependency graph. Furthermore, in the discussion section of the updated version, we acknowledge that the dependency graphs in our dataset are semi-complete, reflecting the inherent challenge of obtaining complete ground-truth graphs in real-world complex systems.
> > > > \
> > > > \
> > > > **Response to: 4. Please add additional baselines compared during rebuttal in the main paper.**
> > > > \
> > > > \
> > > > A: Thank you for your suggestion! We have included the additional baselines in the main paper. Their definitions are detailed in **Section 4.1** (Baseline Definition), and the corresponding experimental results are presented in **Table 3**.
> > > > \
> > > > \
> > > > **Response to: 5. Do you mean that you computed the causal graph separately for each test case? Also, which four cases?**
> > > > \
> > > > \
> > > > Thank you for your question! Yes, we computed the causal graph separately for each failure case, as the RCA task is inherently case-by-case. The four cases refer to the four distinct system failures included in the Product Review dataset.
> > > >
> > > > For each failure case, we computed the metrics individually and then averaged the results across all four cases to ensure consistency and comparability.
> > > >
> > > > We hope this clarifies your concern!  If there is anything unclear in our previous response to any question, please let us know, and we will gladly provide further clarification.

---

> > > > > ### Comment · Reviewer_E7LX · 2024-11-22
> > > > > **Thanks and no further questions.**
> > > > >
> > > > > Thanks authors for the prompt response. I have no further questions.
> > > > > I have increased my score to 5.
> > > > > My last concern is relevance of this work to the iclr main track. Had this been a datasets track I would have strongly championed for the paper. But I would like to depend on other reviewers assessment for this point. I will again update the score if needed.

---

> > > > > > ### Author Response · Authors · 2024-11-22
> > > > > > **Clarifying the Relevance of Our Work to the ICLR Main Track**
> > > > > >
> > > > > > Thank you for your thoughtful feedback and for increasing your score. We would like to address your concern regarding the relevance of this work to the ICLR main track. While ICLR does not have a specific "Datasets and Benchmarks" track like NeurIPS, **the conference does feature a primary area in the main track titled "Datasets and Benchmarks."** This demonstrates that the conference recognizes and values contributions in this area as part of its scope. We hope this clarification highlights the paper’s alignment with the main track and encourages your continued support.
> > > > > >
> > > > > > **We greatly appreciate your time and thoughtful consideration.**

---

> > > > > > > ### Comment · Reviewer_E7LX · 2024-11-26
> > > > > > > **Thank you authors**
> > > > > > >
> > > > > > > Thank you for your active engagement during the rebuttal and for promptly addressing the queries. I have one final request:
> > > > > > >
> > > > > > > While these datasets are valuable for RCD tasks, I also see their utility in temporal causal discovery tasks, an area currently lacking robust datasets. Could you consider releasing the partial graphs as an adjacency matrix or a CSV file, formatted as used in your implementation? Additionally, I would appreciate it if you could release the code for the causal discovery experiments conducted in the paper. This is not urgent, as I understand it might require some time for cleanup.
> > > > > > >
> > > > > > > While I agree that your paper makes a **strong** contribution to the *datasets* aspect, I share the sentiment of other reviewers that the *benchmarking* section needs an appropriate selection of baselines, and a more thorough presentation of the results.
> > > > > > >
> > > > > > > I will retain my score for now.

---

> ### Author Response · Authors · 2024-12-01
> **Thanks Reviewer E7LX for Your Feedback: Response on Baselines, Results, and Data Accessibility**
>
> Thank you for your constructive feedback and for actively engaging with our work during the public discussion phase. We appreciate your thoughtful comments and suggestions.
>
> 1. **Partial Dependency Graphs**:
>   Following your suggestions, we have uploaded the semi-complete dependency graphs in CSV format, which you can access [here](https://drive.google.com/drive/u/4/folders/1mUkgidLaQlfH2Ka8bIq38pQNLcKvyySZ). We will also make the code for the causal discovery experiments available to support reproducibility and further research.
>
> 2. **Selection of Baselines**:
>    Regarding the question about the selection of baselines, we have addressed this in our response to Reviewer UCbX. To summarize briefly, most of the methods we employ—such as CORAL, REASON, MULAN, ϵ-Diagnosis, Nezha, CIRCA, RCD, and Baro—are state-of-the-art algorithms specifically designed for root cause analysis in time-series data. This ensures that the comparative analysis is both meaningful and relevant.  For PC and Notears, while they were not originally designed for time-series data, they have been widely adopted in recent literature for detecting root causes in similar datasets, as shown in:
> - Wang, Lu, Chaoyun Zhang, Ruomeng Ding, Yong Xu, Qihang Chen, Wentao Zou, Qingjun Chen et al. "Root cause analysis for microservice systems via hierarchical reinforcement learning from human feedback." In Proceedings of the 29th ACM SIGKDD Conference on Knowledge Discovery and Data Mining, pp. 5116-5125. 2023.
> - Ikram, Azam, Sarthak Chakraborty, Subrata Mitra, Shiv Saini, Saurabh Bagchi, and Murat Kocaoglu. "Root cause analysis of failures in microservices through causal discovery." Advances in Neural Information Processing Systems 35 (2022): 31158-31170.
> - Zan, Lei. "Causal Discovery from Heterogenous Multivariate Time Series." In Proceedings of the 33rd ACM International Conference on Information and Knowledge Management, pp. 5499-5502. 2024.
> - Yuan Meng, Shenglin Zhang,, Yongqian Sun, Ruru Zhang, Zhilong Hu, Yiyin Zhang, Chenyang Jia, Zhaogang Wang, Dan Pei, “Localizing Failure Root Causes in a Microservice through Causality Inference“. IWQoS 2020.
>
> These references validate their utility and relevance for time-series root cause analysis tasks.
>
> 3. **More Thorough Presentation of Results**:
>    We encourage the reviewer to refer to Section 4.2 of the updated manuscript, where the red-highlighted text provides a detailed explanation of the experimental results, performance distinctions, and underlying reasoning. Quantitative results are presented in Tables 3 and 4, and we have included additional details in Appendix K for further clarity. If there are specific aspects of the results that require further elaboration, we would be happy to address them.
>
> Thank you once again for your valuable feedback and for recognizing the contributions of our work. We remain committed to addressing your concerns and strengthening the impact of this research.

---

### Meta-Review · Area_Chair_ScaX · 2024-12-20

**Metareview:**

The paper contributes
(1) large dataset for root cause analysis
(2) 14 baselines evaluated .

The reviewers feel that if ICLR had a separate Dataset track this would be a sure accept.
The methodological contributions are modest and hence it is not clear on how this paper will stand with other ICLR papers which have more methodological contributions. At this point the paper is, at best, borderline.

**Additional Comments On Reviewer Discussion:**

The points of discussion were
1. Relevance to main track of ICLR
2. Clarification of Baselines
3. Lack of Error bars

The authors responded to all them.

---

### Decision · Program_Chairs · 2025-01-22

Reject